# Polyp Segmentation by Dual-Domain Reasoning: Fuzzy Spatial Control and Frequency Selection

## Abstract

Colorectal cancer (CRC) screening relies on accurate polyp segmentation, yet subtle appearance differences and ambiguous boundaries in colonoscopy images make this task challenging. To overcome these limitations, we propose FSFMamba, a dual-domain fusion network that jointly models boundary uncertainty and frequency structure to improve delineation. In the spatial domain, a Fuzzy Spatial Control Mechanism (FSCM) instantiates an interval type-2 membership to localize uncertainty at boundary bands while preserving stability in homogeneous regions. In the spectral domain, a Frequency Adaptive Selection Mechanism (FASM) performs octave-wise spectral decomposition and applies learnable band-wise weighting to emphasize task-relevant subbands and suppress spurious responses. The two streams are fused by a Mamba-based state-space block that enables long-range, low-latency interactions and pre-norm residual refinement for stable optimization. Extensive experiments show FSFMamba consistently outperforms recent baselines with sharper boundaries, fewer false positives, and strong robustness.

## 1 Introduction

Colorectal cancer (CRC) is the third most common cancer worldwide, accounting for 10% of all cases, and remains the second leading cause of cancer-related deaths. Notably, around 85% of CRC cases arise from adenomatous polyps (Mathews et al., 2021). Early detection and removal can significantly reduce incidence and mortality, achieving a 5-year survival rate of up to 90% (Jiang et al., 2023). Consequently, there is a critical need for automated and reliable polyp segmentation methods to support physicians in accurately identifying polyp regions during diagnosis.

Recent advances in deep learning have significantly improved polyp segmentation performance (Lu et al., 2024; Lijin et al., 2024). However, accurate delineation remains challenging due to the frequent presence of indistinct and ambiguous polyp boundaries (Fig. 1(a) and (b)). To mitigate this, various methods have introduced boundary-aware modules, such as reverse attention (Zhao et al., 2019), balanced attention (Nguyen et al., 2021), and uncertainty-augmented context attention (Kim et al., 2021). Yet, these models often fail in cases involving severe boundary ambiguity. To further enhance boundary modeling, some studies have employed Bayesian estimation frameworks (Djebra et al., 2025; Han et al., 2025) to capture prediction uncertainty. Despite these efforts, effectively distinguishing polyps from fuzzy or overlapping background regions remains an open problem. This raises a key question: **How can uncertainty be effectively modeled to better distinguish polyps from the background under boundary ambiguity?**

Furthermore, our investigation reveals that existing methods primarily focus on single spatial features, which, despite their effectiveness in segmentation, are prone to interference from complex backgrounds due to their reliance on pixel-level information, particularly local intensity and spatial position (Xu et al., 2024b). This limitation hampers their ability to capture global correlations, making it difficult to distinguish subtle variations within polyps and surrounding tissues. Recent studies indicate that frequency features extracted via Fourier Transform (FT) (Qin et al., 2021) or Discrete Cosine Transform (DCT) (Xu et al., 2024b) provide global contextual representations, improving image interpretation and alleviating spatial limitations. However, these methods primarily target high- and low-frequency components, potentially neglecting critical mid-frequency information that

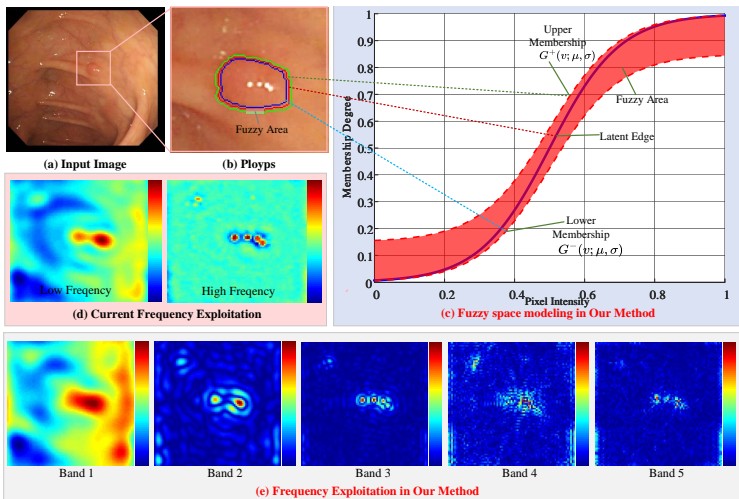

Figure 1: Motivation. (a) is CRC image, while (b) highlights the magnified polyp region with ambiguous boundaries and significant uncertainty. For this, we employ a *Fuzzy Spatial Control Mechanism*, where $G^+(v; \mu, \sigma)$ and $G^-(v; \mu, \sigma)$ define the uncertainty bounds, refining key cues by fuzzy control, as shown in (c). (d) enhances contours and edges by frequency cues, but boundary ambiguity and interference limit its effectiveness. (e) shows the *Frequency Adaptive Selection Mechism*. We split Fourier features into fixed radial octave bands and learn weights to select frequencies, then invert to refine features, mainly strengthening boundary cues.

encodes essential structural details, as shown in Fig. 1 (d). Thus, another important question arises: **How can frequency cues be optimally utilized to extract meaningful information?**

To address the above challenges, we propose FSFMamba, a joint fuzzy spatial-frequency learning framework built on a Mamba backbone for precise polyp segmentation. It tackles both boundary ambiguity and multi-frequency representation. To address *Problem 1*, we design the Fuzzy Spatial Control Mechanism (FSCM), which leverages fuzzy set theory to model boundary uncertainty. By employing second-order membership functions with upper and lower bounds $G^+(v; \mu, \sigma)$ and $G^-(v; \mu, \sigma)$, FSCM adaptively captures edge ambiguity and transition regions (see Fig. 1(c)). To solve *Problem 2*, we propose the Frequency Adaptive Selection Mechanism (FASM), which leverages spectral decomposition to derive subband representations and employs a learnable weighting scheme to selectively amplify discriminative frequency components. As shown in Fig. 1 (e), FASM decomposes Fourier features into fixed octave bands and learns their weights, which are then mapped back to refine spatial predictions. To jointly optimize both domains, we integrate FSCM and FASM into a Dual-Domain Perception Mechanism (D2PM), forming the Frequency Learning and Fuzzy Spatial Control (FLFSC). In a nutshell, the main contributions are listed as:

• To our knowledge, we are the first to employ fuzzy control to model ambiguous boundary uncertainty and exploit multi-frequency bands to capture subtle variations, thereby enhancing polyp segmentation.

• We propose the joint fuzzy spatial-frequency learning Mamba network (FSFMamba) integrating multi-level FLFSC for learning of fuzzy spatial features and frequency cues. Each FLFSC integrates: ($i$) FSCM for capturing boundary uncertainty and adapts to membership variability, ($ii$) FASM for modeling intrinsic correlations across frequency bands and learns dependencies among frequency components, and ($iii$) D2PM for combining both to enhance polyp segmentation.

• Extensive comparative experiments on public datasets demonstrate that our method consistently provides robust segmentation performance across various challenging scenarios.

## 2 RELATED WORKS

**Polyp Segmentation.** Recent polyp segmentation spans CNN, Transformer, and hybrid designs. CNNs with VGG (Vedaldi & Zisserman, 2016) and ResNet (Koonce, 2021), enhanced by attention (Kim et al., 2021; Nguyen et al., 2021; Zhao et al., 2019), interaction (Zhang et al., 2022b), edge-

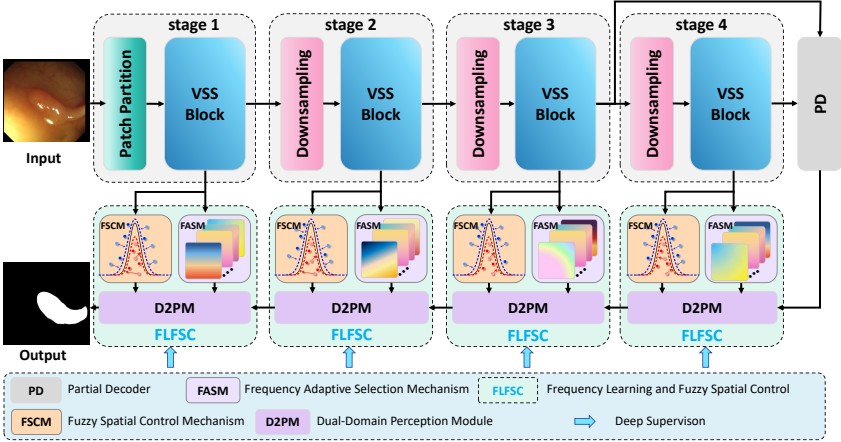

Figure 2: Overview of the proposed FSFMamba framework for polyp segmentation. Among these, FSCM resolves boundaries by Gaussian fuzzy regression, FASM filters critical frequency sub-bands, and D2PM fuses fuzzy spatial feature and spectral feature by visual state-space operators.

aware (Su et al., 2023), and multi-scale modules (Ji et al., 2024a), capture fine detail but lack global context. Transformers improve long-range reasoning via self-attention, as in ViT (Chen et al., 2021), Swin-Unet (Cao et al., 2023), and UNetFormer (Wang et al., 2022), often with higher compute. Hybrids such as TransFuse (Zhang et al., 2021), PolypPVT (Bo et al., 2023), and SSFormer (Shi et al., 2022) integrate global and local cues. Foundation models (SAM (Wei et al., 2024b), SAM2-UNet (Xiong et al., 2024)) and VMamba (Liu et al., 2024b) extend efficient global modeling. Following this, we adopt Mamba as the backbone for efficient feature extraction.

**Fuzzy Learning.** Deep learning excels at large-scale, task-driven feature extraction, but deterministic models handle uncertainty poorly. To mitigate this, fuzzy logic has been integrated into neural networks (Luo et al., 2023; Mohammadzadeh et al., 2023). Huang *et al.* (Xie et al., 2021) use fuzzy memberships to quantify pixel-level ambiguity in segmentation. Wang *et al.* (Wang et al., 2023) map images into a fuzzy domain for rule-based reasoning and fuse the result with convolutional features. Wei *et al.* (Wei et al., 2024a) detect boundary pixels via local variation with fuzzy awareness. We instead adopt a Gaussian-regressed interval type-2 membership that converts rigid constraints into elastic spatial boundaries, improving structural adaptability and reducing uncertainty.

**Frequency Learning.** Frequency analysis, central to signal processing (Pitas, 2000), is increasingly applied in vision to guide optimization (Yin et al., 2019), enable non-local feature learning (Huang et al., 2023), and support domain-generalizable representations (Lin et al., 2023b). In polyp segmentation, Ren *et al.* (Ren et al., 2024) leverage high-frequency cues with a local–nonlocal Transformer. Recent work integrates spatial and frequency cues: Yue *et al.* (Yue et al., 2024) fuse them via interaction learning, and Li *et al.* (Li et al., 2024) apply parameterized frequency modulation to refine styles and enhance lesions. Although frequency–spatial fusion is common, polyp noise and texture ambiguity undermine feature reliability. We propose fuzzy-controlled spatial optimization with adaptive frequency selection to jointly strengthen representations across domains.

## 3 METHODOLOGY

### 3.1 OVERVIEW

The framework of our FSFMamba is shown in Fig. 2, in which Mamba is adopted as the backbone (see Appendix A.2.1) owing to its demonstrated ability to capture long-range dependencies with lower computational overhead compared to full-attention Transformers. Given an input image $I \in \mathbb{R}^{H \times W \times 3}$, VMamba (Liu et al., 2024b) extracts multi-scale features $\{F_i\}_{i=1}^4$ with dimensions $\frac{H}{2^{i+1}} \times \frac{W}{2^{i+1}} \times C_i$. These are sequentially processed by Frequency Learning and Fuzzy Spatial Control (FLFSCs), which integrate a **fuzzy spatial control** mechanism (FSCM), a **frequency adaptive selection** mechanism (FASM), and a dual-domain perception module (D2PM) to refine feature

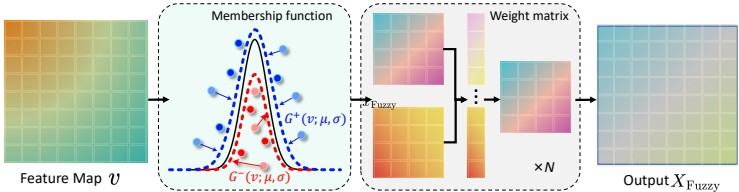

Figure 3: The illustration of FSCM constrained feature map.

representation. A partial decoder (PD; see Appendix A.2.2) provides semantic priors to the deepest FLFSC, while each subsequent FLFSC is hierarchically guided by the preceding output.

## 3.2 FUZZY SPATIAL CONTROL

In polyp segmentation, sharp boundary labels are unreliable because polyps and mucosa often share highly similar visual traits. To better capture this ambiguity, fuzzy logic assigns graded memberships to polyp and background, thereby enabling smooth boundary transitions and uncertainty-aware supervision. Building on this principle, FSCM explicitly regularizes boundary features to mitigate annotation brittleness, as illustrated in Fig. 3. By enforcing fuzzy learning, boundaries become smoother and the network gains stronger discrimination ability, akin to Gaussian fuzzy mechanisms (Liu, 2018) that assign membership based on neighboring pixel positions:

$$G_i(v; \mu, \sigma) = \frac{\lambda}{\sqrt{2\pi}\sigma} \mathrm{e}^{-\frac{(v_i - \mu)^2 + \frac{1}{\rho(R_i)} \sum_{i \in R_{i'}} (v_{i'} - \mu)^2}{2\sigma^2}},$$ (1)

where $G_i(\cdot)$ represents the membership function, $i$ is the pixel index, $\lambda$ is the model coefficient, and $v_i$ is the deep feature activation at pixel $i$ from an intermediate feature map. $\mu$ and $\sigma$ are the mean and standard deviation of the pixel values, respectively. $R_i$ is the 8-neighbor set in a $3 \times 3$ window centered at pixel $i$, $R_i'$ is derived from the signed distance map of predicted boundary logits within a local neighborhood, and $\rho(R_i)$ denotes the neighborhood scale normalization over $R_i$, implemented as a differentiable local standard deviation:

$$\rho(R_i) = \sqrt{\frac{1}{|R_i|} \sum_{u \in R_i} (u - \bar{u})^2 + \varepsilon}.$$ (2)

This term estimates local feature variation and adaptively normalizes the membership response. To account for non-Gaussian deviations, we introduce asymmetric fuzzy membership functions $G_i^+$ and $G_i^-$, representing interval-based pixel-wise confidence. $G_i^+$ captures positive deviations, while $G_i^-$ encodes negative deviations, jointly modeling uncertainty with flexible context-aware constraints. The upper membership function $G_i^+(\cdot)$ is defined as:

$$G_i^+(v; \mu, \sigma) = \begin{cases} \frac{\lambda}{\sqrt{2\pi}\sigma} \mathrm{e}^{-\frac{(v_i - \mu^-)^2 + \frac{1}{\rho(R_i)} \sum_{i \in R_{i'}} (v_{i'} - \mu^-)^2}{2\sigma^2}}, & \text{if } v_i < \mu^- \\ \frac{\lambda}{\sqrt{2\pi}\sigma} \mathrm{e}^{-\frac{(v_i - \mu^+)^2 + \frac{1}{\rho(R_i)} \sum_{i \in R_{i'}} (v_{i'} - \mu^+)^2}{2\sigma^2}}, & \text{if } v_i > \mu^+ \end{cases}$$ (3)

In a similar way, the lower membership function, $G_i^-(\cdot)$, expressed as:

$$G_i^-(v; \mu, \sigma) = \begin{cases} \frac{\lambda}{\sqrt{2\pi}\sigma} \mathrm{e}^{-\frac{(v_i - \mu^+)^2 + \frac{1}{\rho(R_i)} \sum_{i \in R_{i'}} (v_{i'} - \mu^+)^2}{2\sigma^2}}, & \text{if } v_i \leq \frac{\mu^- + \mu^+}{2} \\ \frac{\lambda}{\sqrt{2\pi}\sigma} \mathrm{e}^{-\frac{(v_i - \mu^-)^2 + \frac{1}{\rho(R_i)} \sum_{i \in R_{i'}} (v_{i'} - \mu^-)^2}{2\sigma^2}}, & \text{if } v_i > \frac{\mu^- + \mu^+}{2} \end{cases}$$ (4)

where the mean value $\mu$ is adjusted to the interval $[\mu^-, \mu^+]$, with $\mu^-$ and $\mu^+$ representing the mean values at the left and right boundaries of the interval, are respectively calculated as:

$$\mu^- = \mu - \xi \times \sigma, \ \mu^+ = \mu + \xi \times \sigma,$$ (5)

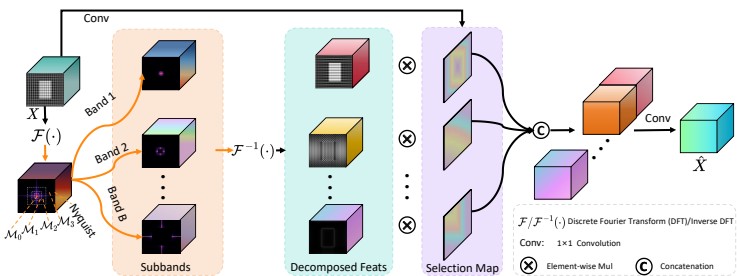

Figure 4: The illustration of FASM.

where $\xi$ is the interval adjustment factor for mean deviation. Under a Gaussian membership, 99.7% of mass falls in $[\mu-3\sigma, \mu+3\sigma]$. We control uncertainty with $\xi \in [0,3]$, shifting the lower/upper memberships to form adaptive constraint intervals. The fuzzy feature is $X_{\text{fuzzy}} = \sum_{i=1}^{R} G_i^{+|-}(v; \mu, \sigma) v_i$ and $\xi$ adjusts $G^+/G^-$ weight kernel responses. Detailed theoretical analysis is provided in Appendix A.3.

## 3.3 FREQUENCY ADAPTIVE SELECTION

Standard CNNs are dominated by local inductive bias. Without an explicit mechanism to regulate spectral content, they tend to either overfit high-frequency noise or under-exploit mid-range texture cues that are critical for polyp delineation, especially under blur and low contrast. Motivated by this, we introduce the Frequency Adaptive Selection Module (FASM) to explicitly factor and reweight frequency components before dual-domain fusion. As illustrated in Fig. 4, FASM suppresses unstable high-frequency responses while preserving informative mid-frequency structures, thereby yielding a more frequency-balanced representation.

Given a channel feature map $X_c \in \mathbb{R}^{H \times W}$ with $c \in \{1, 2, \ldots, C_i\}$, we transform it into the Fourier domain via the discrete Fourier transform (DFT),

$$X_{\mathcal{F},c}(u,v) = \frac{1}{HW} \sum_{h=0}^{H-1} \sum_{w=0}^{W-1} X_c(h,w) e^{-2\pi j(uh+vw)}, \tag{6}$$

where $(h, w)$ denotes spatial coordinates and $(u, v)$ denotes normalized frequencies along height and width. After shifting the spectrum to center low-frequency components, $u$ and $v$ range over $\{-\frac{H}{2}, \ldots, \frac{H}{2} - 1\}$ and $\{-\frac{W}{2}, \ldots, \frac{W}{2} - 1\}$, respectively. Frequencies beyond the Nyquist limit $\mathcal{H}_{D_+} = \{(u,v) \mid |u| > \frac{1}{2D} \text{ or } |v| > \frac{1}{2D}\}$ cannot be faithfully represented, which effectively bounds the usable bandwidth.

Different from wavelet or DWT-based decompositions that yield predefined multi-scale sub-bands, we adopt a simple and interpretable octave-wise Fourier partition. Specifically, we decompose $X_{\mathcal{F},c}$ into multiple fixed sub-bands using binary masks,

$$X_{b,c} = \mathcal{F}^{-1}\big(\mathcal{M}_{b,c} X_{\mathcal{F},c}\big), \tag{7}$$

where $\mathcal{F}^{-1}$ is the inverse DFT and $\mathcal{M}_{b,c}$ is defined by fixed octave thresholds,

$$\mathcal{M}_{b,c}(u,v) = \begin{cases} 1, & \text{if } \phi_b \leq \max(|u|, |v|) < \phi_{b+1}, \\ 0, & \text{otherwise.} \end{cases} \tag{8}$$

where $\{\phi_b\}_{b=0}^{B}$ are predetermined frequency boundaries, yielding an octave-wise split of the normalized spectrum. In all experiments, we use four bands $[0, \frac{1}{16})$, $[\frac{1}{16}, \frac{1}{8})$, $[\frac{1}{8}, \frac{1}{4})$, and $[\frac{1}{4}, \frac{1}{2}]$.

Crucially, FASM does *not* learn band boundaries. Instead, it learns spatially varying selection maps $A_b$ to adaptively reweight fixed bands,

$$\widehat{X} = \text{Conv}\big(\text{Cat}\big[A_0 \odot X_0, A_1 \odot X_1, \ldots, A_{B-1} \odot X_{B-1}\big]\big), \tag{9}$$

where $\widehat{X}$ is the frequency-balanced representation. The selection map $A_b \in \mathbb{R}^{H \times W \times C_i}$ for band $b$ is predicted from the input features via $A_b = \text{Conv}_b(X)$. The operator $\odot$ denotes element-wise

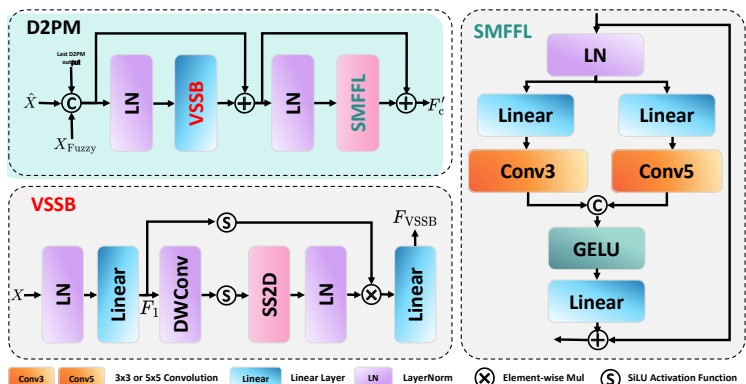

Figure 5: The illustration of D2PM.

multiplication, $\mathrm{Cat}[\cdot]$ is channel-wise concatenation, and $\mathrm{Conv}(\cdot)$ is a $1 \times 1$ convolution for fusing band-specific responses. This design yields a fixed, octave-wise spectral basis for interpretability, while allowing the model to emphasize the most informative bands in a content-aware manner. Detailed theoretical analysis is provided in Appendix A.4.

### 3.4 DUAL-DOMAIN PERCEPTION

The D2PM is proposed to fuse fuzzy spatial and frequency cues to capture cross-scale mixed information and long-range dependencies. As shown in Fig. 5, the D2PM consists of two stages: the input is first normalized and processed by the Visual State Space Block (VSSB) to capture global context, and then passed through the Scale-Mixed Feed-Forward Layer (SMFFL) to extract multi-scale features with residual connections. Formally:

$$F_{\mathrm{cor}} = \mathrm{Cat}(\hat{X}, X_{\mathrm{fuzzy}}, F_{\mathrm{c}}'^{(k-1)}), F_{\mathrm{cor}}' = F_{\mathrm{cor}} + \mathrm{VSSB}(\mathrm{LN}(F_{\mathrm{cor}})), F_{\mathrm{c}}'^{(k)} = F_{\mathrm{cor}}' + \mathcal{S}(\mathrm{LN}(F_{\mathrm{cor}}')), \tag{10}$$

where $\mathrm{LN}(\cdot)$ refers to the layer normalization and $\mathcal{S}(\cdot)$ is the SMFFL.

**VSSB.** Fig. 5 provides the structure of the VSSB. The input features $X$ are first subjected to layer normalization, followed by processing by a linear layer, resulting in the two separate streams as:

$$F_1 = \mathrm{Fc}(\mathrm{LN}(X)), F_1' = \mathrm{LN}(\mathrm{SS2D}(\mathrm{SiLU}(\mathrm{DW}(F_1)))), F_{\mathrm{VSSB}} = \mathrm{Fc}(F_1' \otimes \mathrm{SiLU}(F_1)), \tag{11}$$

where $\mathrm{DW}(\cdot)$ refers to depth-wise separable convolution, $\mathrm{SiLU}(\cdot)$ represents SiLU activation function, $\mathrm{Fc}(\cdot)$ is linear layer, $\mathrm{SS2D}(\cdot)$ denotes 2D selective scanning, and $F_{\mathrm{VSSB}}$ is the output of VSSB.

**SMFFL.** As shown in Fig. 5, SMFFL adopts a dual-branch, multi-scale structure. The input feature $F_{\mathrm{X}}$ is normalized and split into two branches with linear projection and $3 \times 3$ or $5 \times 5$ convolutions to capture multi-scale features in a low-dimensional space. After GELU activation and up-projection, residual connections are added to support gradient flow. The process is defined as:

$$F_{\mathrm{X}}' = \mathrm{LN}(F_{\mathrm{X}}), F_{\mathrm{X}1} = f_3(\mathrm{Fc}(F_{\mathrm{X}}')), F_{\mathrm{X}2} = f_5(\mathrm{Fc}(F_{\mathrm{X}}')), F_{\mathrm{S}} = \mathrm{Fc}(\sigma_{\mathrm{G}}(\mathrm{Cat}(F_{\mathrm{X}1}, F_{\mathrm{X}2}))) + F_{\mathrm{X}}, \tag{12}$$

where $f_x(\cdot)$ denotes the standard convolution operation of size $x \times x$, $\sigma_{\mathrm{G}}(\cdot)$ represents the GELU activation function, and $\oplus$ indicates element-wise addition.

### 3.5 LOSS FUNCTIONS

The loss function utilized in this study is formulated as $\mathcal{L} = \mathcal{L}_{w\mathrm{IoU}} + \mathcal{L}_{w\mathrm{BCE}}$ (Jun Wei, 2020), where $\mathcal{L}_{w\mathrm{IoU}}$ represents the weighted intersection over union (IoU) loss and $\mathcal{L}_{w\mathrm{BCE}}$ denotes the weighted binary cross-entropy (BCE) loss. We implement the $\mathcal{L}$ to facilitate deep supervision across the four outputs $\{t_i, i = 1, 2, 3, 4\}$. Thus, the total loss is $\mathcal{L}_{\mathrm{total}} = \sum_{i=1}^{i=4} \mathcal{L}(t_i^{up}, G)$.

Table 1: Quantitative comparison of our method against other models on two domain-specific datasets: CVC-ClinicDB and Kvasir-SEG. Best results are in red.

| Method | CVC-ClinicDB | | | | | | Kvasir-SEG | | | | | |
|---|---|---|---|---|---|---|---|---|---|---|---|---|
| | mDSC ↑ | mIoU ↑ | wFm ↑ | Sm ↑ | MAE ↓ | maxEm ↑ | mDSC ↑ | mIoU ↑ | wFm ↑ | Sm ↑ | MAE ↓ | maxEm ↑ |
| CaraNet | 0.9045 | 0.8480 | 0.8943 | 0.9379 | 0.0124 | 0.9729 | 0.9046 | 0.8465 | 0.8869 | 0.9188 | 0.0273 | 0.9653 |
| DCRNet | 0.8962 | 0.8440 | 0.8902 | 0.9337 | 0.0101 | 0.9779 | 0.8864 | 0.8248 | 0.8681 | 0.9106 | 0.0354 | 0.9412 |
| SSFormer | 0.9160 | 0.8730 | 0.9240 | 0.9370 | 0.0070 | 0.9847 | 0.9250 | 0.8780 | 0.9210 | 0.9310 | 0.0170 | 0.9643 |
| HSNet | 0.9476 | 0.9050 | 0.9508 | 0.9543 | 0.0055 | 0.9933 | 0.9258 | 0.8771 | 0.9177 | 0.9268 | 0.0234 | 0.9639 |
| CFANet | 0.9325 | 0.8828 | 0.9241 | 0.9507 | 0.0068 | 0.9893 | 0.9147 | 0.8615 | 0.9029 | 0.9240 | 0.0229 | 0.9623 |
| FeDNet | 0.9304 | 0.8846 | 0.9284 | 0.9501 | 0.0069 | 0.9817 | 0.9242 | 0.8761 | 0.9180 | 0.9329 | 0.0212 | 0.9664 |
| PolypPVT | 0.9368 | 0.8894 | 0.9355 | 0.9500 | 0.0064 | 0.9891 | 0.9174 | 0.8642 | 0.9105 | 0.9251 | 0.0228 | 0.9617 |
| MSCAF-Net | 0.9261 | 0.8786 | 0.9222 | 0.9503 | 0.0064 | 0.9818 | 0.9113 | 0.8565 | 0.9026 | 0.9218 | 0.0248 | 0.9636 |
| CAFE-Net | 0.9326 | 0.8889 | 0.9316 | 0.9549 | 0.0064 | 0.9816 | 0.9210 | 0.8742 | 0.9145 | 0.9319 | 0.0211 | 0.9700 |
| PGCF | 0.9397 | 0.8938 | 0.9396 | 0.9520 | 0.0057 | 0.9925 | 0.9117 | 0.8622 | 0.9049 | 0.9214 | 0.0241 | 0.9610 |
| CTNet | 0.9355 | 0.8875 | 0.9344 | 0.9529 | 0.0063 | 0.9876 | 0.9171 | 0.8628 | 0.9100 | 0.9280 | 0.0232 | 0.9640 |
| SAM2UNet | 0.9041 | 0.8535 | 0.8983 | 0.9464 | 0.0097 | 0.9683 | 0.9241 | 0.8760 | 0.9169 | 0.9394 | 0.0198 | 0.9727 |
| DBG-Net | 0.9047 | 0.8571 | 0.8982 | 0.9367 | 0.0079 | 0.9684 | 0.9152 | 0.8626 | 0.9062 | 0.9196 | 0.0253 | 0.9637 |
| **Ours** | 0.9522 | 0.9112 | 0.9510 | 0.9600 | 0.0053 | 0.9945 | 0.9358 | 0.8951 | 0.9331 | 0.9403 | 0.0176 | 0.9705 |

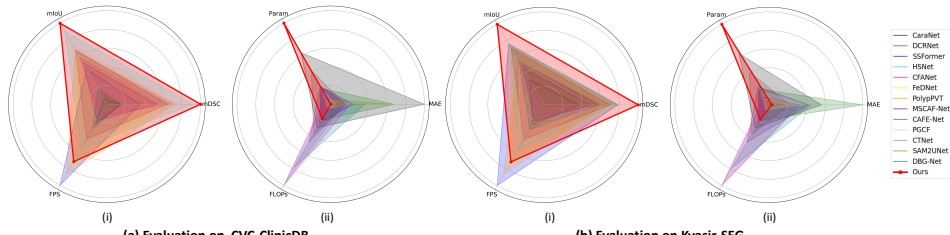

Figure 6: Complexity comparison on CVC-ClinicDB and Kvasir-SEG. Metrics are split by optimization objective for clarity: (i) higher-is-better metrics and (ii) lower-is-better metrics.

# 4 EXPERIMENTS

## 4.1 SETUPS

**Datasets.** Following the experimental setups in (Bo et al., 2023), we systematically evaluate the performance of our method across five prominent public datasets focused on polyp segmentation: CVC-300, CVC-ClinicDB, Kvasir-SEG, CVC-ColonDB, and ETIS.

**Compared Models and Evaluation Metrics.** We compare our method against 13 public polyp segmentation models, including CaraNet (Lou et al., 2021), DCRNet (Yin et al., 2022), SSFormer (Shi et al., 2022), HSNet (Zhang et al., 2022b), CFANet (Zhang & Yan, 2023), FeDNet (Su et al., 2023), Polyp-PVT (Bo et al., 2023), MSCAF-Net (Liu et al., 2023), CAFENet (Liu et al., 2024a), PGCF (Ji et al., 2024b), CTNet (Xiao et al., 2024), SAM2UNet (Xiong et al., 2024), and DBG-Net (Zhai et al., 2024). For fairness, we adopt their official codes and evaluate all models under identical training and testing settings. Each method is rigorously assessed by 6 widely recognized metrics (Bo et al., 2023): mean Dice Similarity Coefficient (mDSC), mean Intersection over Union (mIoU), Weighted F-measure (wFm), S-measure (Sm), max E-measure (maxEm), and Mean Absolute Error (MAE).

**Implementation Details.** Our method is developed on the PyTorch framework, with the VMamba-Small model pretrained on ImageNet as the backbone. To account for variations in polyp image sizes, a multi-scale strategy of $\{0.75, 1, 1.25\}$ is employed instead of conventional data augmentation. Input images are resized to $352 \times 352$ pixels, with a mini-batch size of 16, and training is 100 epochs. The AdamW optimizer is used to fine-tune the model parameters, with a learning rate and weight decay both set to 1e–4. The training process, executed on an NVIDIA A5000 GPU.

## 4.2 COMPARISON WITH STATE-OF-THE-ARTS

**Quantitative Analysis of Learning Ability.** We conduct a quantitative comparison on two clinically relevant benchmarks, CVC-ClinicDB and Kvasir-SEG (Table 1). Our method consistently outper-

Table 2: Quantitative comparison of our method against other models across three out-of-domain datasets: CVC-300, CVC-ColonDB, and ETIS. Best results are in red.

| Method | CVC-300 | | | | | | CVC-ColonDB | | | | | | ETIS | | | | | |
|---|---|---|---|---|---|---|---|---|---|---|---|---|---|---|---|---|---|---|
| | mDSC | mIoU | wFm | Sm | MAE | maxEm | mDSC | mIoU | wFm | Sm | MAE | maxEm | mDSC | mIoU | wFm | Sm | MAE | maxEm |
| CaraNet | 0.8809 | 0.8109 | 0.8499 | 0.9298 | 0.0096 | 0.9820 | 0.7384 | 0.6558 | 0.7065 | 0.8289 | 0.0454 | 0.8725 | 0.7293 | 0.6444 | 0.6734 | 0.8488 | 0.0159 | 0.9126 |
| DCRNet | 0.8565 | 0.7883 | 0.8303 | 0.9216 | 0.0101 | 0.9597 | 0.7040 | 0.6315 | 0.6839 | 0.8211 | 0.0516 | 0.8480 | 0.5557 | 0.4960 | 0.5063 | 0.7356 | 0.0958 | 0.7730 |
| SSFormer | 0.8870 | 0.8210 | 0.8690 | 0.9290 | 0.0070 | 0.9751 | 0.7720 | 0.6970 | 0.7660 | 0.8440 | 0.0170 | 0.9235 | 0.7670 | 0.6980 | 0.7360 | 0.8630 | 0.0160 | 0.9132 |
| HSNet | 0.9027 | 0.8393 | 0.8868 | 0.9375 | 0.0067 | 0.9750 | 0.8099 | 0.7347 | 0.7955 | 0.8679 | 0.0324 | 0.9146 | 0.8079 | 0.7335 | 0.7775 | 0.8821 | 0.0211 | 0.9090 |
| CFANet | 0.8933 | 0.8269 | 0.8746 | 0.9383 | 0.0080 | 0.9781 | 0.7426 | 0.6649 | 0.7281 | 0.8351 | 0.0388 | 0.8976 | 0.7325 | 0.6549 | 0.6930 | 0.8455 | 0.0143 | 0.8920 |
| FeDNet | 0.9106 | 0.8485 | 0.8971 | 0.9461 | 0.0057 | 0.9854 | 0.8235 | 0.7443 | 0.8089 | 0.8781 | 0.0295 | 0.9219 | 0.8104 | 0.7335 | 0.7729 | 0.8916 | 0.0156 | 0.9414 |
| PolypPVT | 0.9001 | 0.8332 | 0.8835 | 0.9349 | 0.0066 | 0.9812 | 0.8083 | 0.7273 | 0.795 | 0.8654 | 0.0311 | 0.9190 | 0.7869 | 0.7058 | 0.7498 | 0.8709 | 0.0130 | 0.9098 |
| MSCAF-Net | 0.9022 | 0.8362 | 0.8842 | 0.9417 | 0.0061 | 0.9805 | 0.7902 | 0.7109 | 0.7691 | 0.8596 | 0.0313 | 0.9033 | 0.7745 | 0.6977 | 0.7342 | 0.8660 | 0.0165 | 0.9069 |
| CAFE-Net | 0.8867 | 0.8151 | 0.8618 | 0.9304 | 0.0079 | 0.9747 | 0.8181 | 0.7387 | 0.7994 | 0.8760 | 0.0270 | 0.9182 | 0.8275 | 0.7485 | 0.7874 | 0.8996 | 0.0122 | 0.9375 |
| PGCF | 0.8955 | 0.8272 | 0.8732 | 0.9351 | 0.0073 | 0.9762 | 0.8158 | 0.7376 | 0.8013 | 0.8729 | 0.0271 | 0.9233 | 0.7619 | 0.6861 | 0.7290 | 0.8578 | 0.0173 | 0.8862 |
| CTNet | 0.9082 | 0.8437 | 0.8943 | 0.9435 | 0.0058 | 0.9822 | 0.8127 | 0.7336 | 0.8007 | 0.8749 | 0.0272 | 0.9195 | 0.8098 | 0.7337 | 0.7764 | 0.8865 | 0.0139 | 0.9205 |
| SAM2UNet | 0.8901 | 0.8237 | 0.9220 | 0.9579 | 0.0042 | 0.9866 | 0.8048 | 0.7279 | 0.7887 | 0.8773 | 0.0282 | 0.9141 | 0.7930 | 0.7201 | 0.7579 | 0.8815 | 0.0178 | 0.9113 |
| DBG-Net | 0.9019 | 0.8367 | 0.8834 | 0.9400 | 0.0052 | 0.9797 | 0.7971 | 0.7227 | 0.7807 | 0.8698 | 0.0282 | 0.9120 | 0.7521 | 0.6813 | 0.7186 | 0.8604 | 0.0142 | 0.9098 |
| **Ours** | 0.9171 | 0.8674 | 0.9076 | 0.9533 | 0.0056 | 0.9879 | 0.8912 | 0.8347 | 0.8828 | 0.9252 | 0.0158 | 0.9623 | 0.8560 | 0.7939 | 0.8361 | 0.9162 | 0.0076 | 0.9537 |

forms existing approaches across all metrics. On CVC-ClinicDB, it ranks first on mDSC (0.9522), mIoU (0.9112), wFm (0.9510), Sm (0.9600), and maxEm (0.9945), while attaining the lowest MAE (0.0053). The same pattern holds on Kvasir-SEG, where our model leads on core metrics, including mDSC (0.9358) and mIoU (0.8951), evidencing strong generalization. Although SAM2UNet has a marginally higher maxEm (0.9727) on Kvasir-SEG, our approach delivers a more balanced overall profile. The Fig. 6 shows our method has mid-range FLOPs and parameters with high FPS, offering a favorable accuracy–efficiency trade-off.

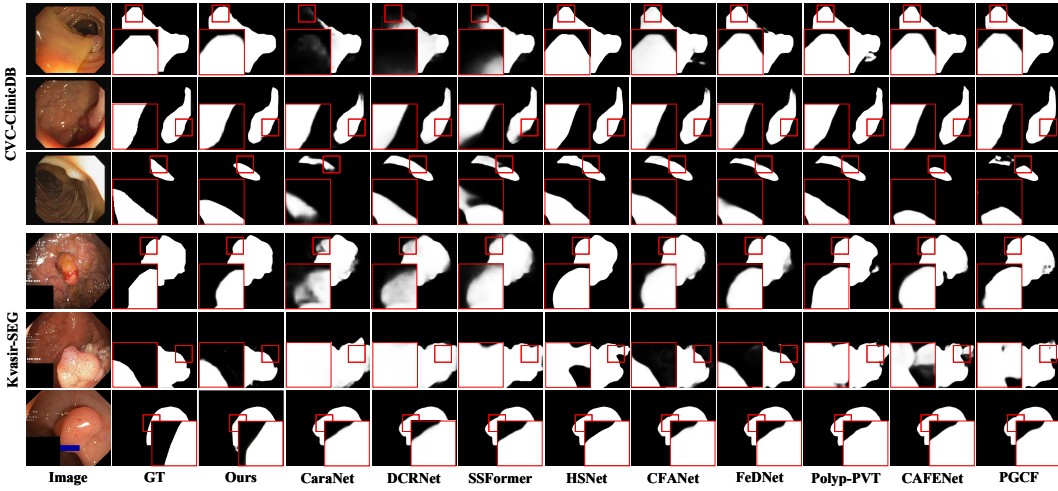

Figure 7: Visualization comparisons on CVC-ClinicDB and Kvasir-SEG datasets.

**Quantitative Analysis of Generalization Ability.** We evaluate out-of-domain performance on CVC-300, CVC-ColonDB, and ETIS (Table 2). Our method consistently surpasses comparison models on most metrics. On CVC-300, it achieves the best mDSC of 0.9171, mIoU of 0.8674, and maxEm of 0.9879. On CVC-ColonDB, where image quality is low and structural variability is high, it reaches the top mDSC of 0.8912 and mIoU of 0.8347. On ETIS, which contains low-resolution and small-scale polyps, it again leads across metrics, including mDSC of 0.8560, mIoU of 0.7939, and Sm of 0.9162. While SAM2UNet peaks on a few isolated metrics, our model provides a more balanced profile, evidencing robust generalization under domain shift.

**Qualitative Analysis.** As shown in Fig. 7, baseline methods such as CaraNet, DCRNet, and SSFormer often miss accurate polyp boundaries under complex shapes and low contrast, yielding fragmented or incomplete masks. More recent models improve overall detection but frequently over-segment ambiguous regions, introducing false positives. In contrast, HSNet, PGCF, and our method achieve

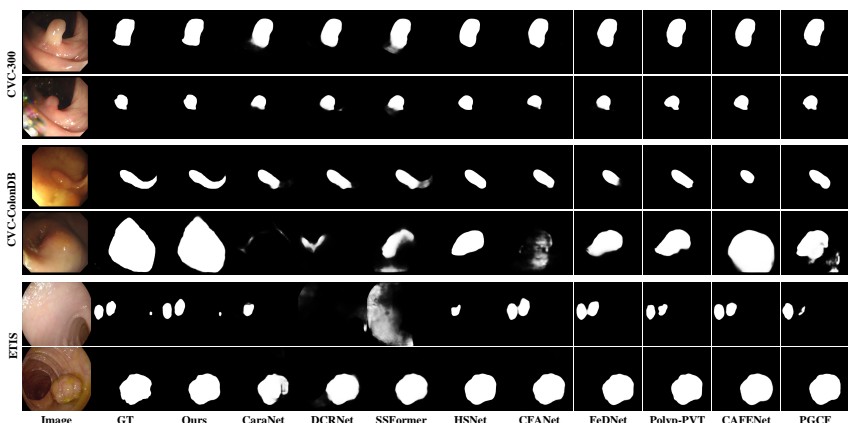

Figure 8: Visualization comparisons on CVC-300, CVC-ColonDB, and ETIS datasets.

Table 3: Ablation analysis of the components in our method. Best results are in **bold**.

| No. | Settings | | | CVC-ClinicDB | | Kvasir-SEG | |
|---|---|---|---|---|---|---|---|
| | baseline | PD | FLFSC | mDSC | mIoU | mDSC | mIoU |
| #1 | ✓ | ✗ | ✗ | 0.8623 | 0.8073 | 0.8537 | 0.8012 |
| #2 | ✓ | ✓ | ✗ | 0.8951 | 0.8255 | 0.8879 | 0.8225 |
| #3 | ✓ | ✗ | ✓ | 0.9312 | 0.8843 | 0.9231 | 0.8754 |
| #4 | ✓ | ✓ | ✓ | **0.9522** | **0.9112** | **0.9358** | **0.8951** |
| | FSCM | FASM | D2PM | mDSC | mIoU | mDSC | mIoU |
| #5 | ✓ | ✗ | ✗ | 0.9158 | 0.8601 | 0.9089 | 0.8524 |
| #6 | ✓ | ✓ | ✗ | 0.9328 | 0.8846 | 0.9214 | 0.8748 |
| #7 | ✓ | ✗ | ✓ | 0.9334 | 0.8873 | 0.9218 | 0.8716 |

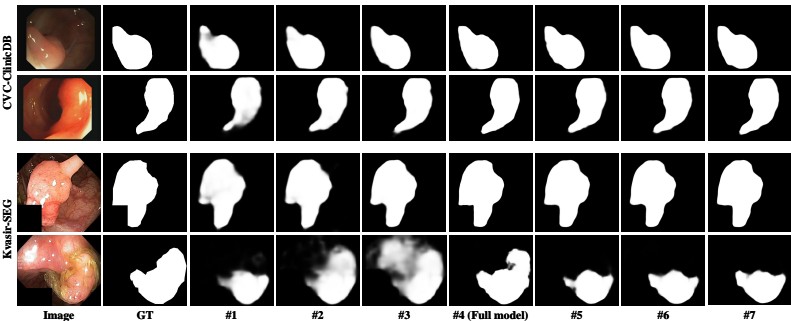

Figure 9: Ablation results on different configurations.

stronger structural alignment with the ground truth. Notably, our approach better preserves fine boundaries while suppressing background noise, reflecting superior spatial awareness and feature discrimination. Fig. 8 further confirms this advantage on CVC-300, CVC-ColonDB, and ETIS, demonstrating robust generalization under domain shifts and challenging clinical conditions.

### 4.3 ABLATION STUDY

**The Effect of Each Component.** Table 3 reports the ablation study of FSFMamba. Starting from the "Vmamba–FPN" baseline (#1), we individually introduce the PD (#2) and FLFSC (#3) components, each yielding clear gains. Their joint integration (#4) further amplifies performance, achieving the best results. Fig. 9 shows that adding PD and FLFSC progressively sharpens predictions toward GT boundaries, confirming their complementarity and the effect of our method.

Figure 10: Octave-wise band-selection maps $A_b$ learned by FASM. Mid-frequency maps (2/3) highlight polyp contours, while low- (1) and high-frequency (4) maps mainly respond to smooth background and noisy details, respectively.

Table 4: Type-2 vs. Type-1 FSCM ablation

| Dataset | Model | mDice ↑ | mIoU ↑ | MAE ↓ |
|---------|-------|---------|--------|-------|
| CVC-ClinicDB | Type-2 FSCM (Full) | 0.9522 | 0.9091 | 0.0068 |
|  | Type-1 FSCM baseline | 0.9417 | 0.8943 | 0.0081 |
| Kvasir-SEG | Type-2 FSCM (Full) | 0.9358 | 0.8896 | 0.0089 |
|  | Type-1 FSCM baseline | 0.9246 | 0.8729 | 0.0103 |
| CVC-ColonDB | Type-2 FSCM (Full) | 0.8820 | 0.8012 | 0.0315 |
|  | Type-1 FSCM baseline | 0.8613 | 0.7784 | 0.0352 |
| ETIS | Type-2 FSCM (Full) | 0.8325 | 0.7420 | 0.0410 |
|  | Type-1 FSCM baseline | 0.8041 | 0.7118 | 0.0461 |

**The Effect of Sub-modules within FLFSC.** To assess the contribution of the FLFSC, we decompose it into FSCM, FASM, and D2PM. As shown in Table 3, FSCM (#5) handles fuzzy boundaries, FASM (#6) selects frequency cues to enrich structural detail, and D2PM (#7) fuses both outputs. Each part provides measurable gains, while the full FLFSC (#4) achieves the best performance. These results confirm that combining spatial and frequency cues is critical for accurate polyp delineation.

**Evidence of Critical Mid-Frequency Selection.** To make FASM's frequency preference explicit, we visualize the learned octave-wise band-selection maps $A_b$ for four bands. Fig. 10 shows that the mid-frequency maps $(A_2, A_3)$ form contiguous high-response belts along polyp contours and thin structures, while the low-frequency map $A_1$ mainly responds to smooth background and coarse illumination and the high-frequency map $A_4$ is sparsely activated in noisy or specular regions. This confirming that FASM prioritizes mid-frequency cues for subtle boundary delineation over extreme low or high frequencies.

**Type-2 vs Type-1 FSCM Ablation Analysis.** We build a Type-1 fuzzy baseline by collapsing FSCM's upper and lower memberships into a single Gaussian membership with the same parameterization, while keeping the backbone, losses, and training protocol identical. As shown in Table 4, the interval Type-2 FSCM consistently improves mDice/mIoU and reduces MAE across all datasets, with larger gains on boundary-ambiguous and out-of-domain sets (*e.g.*, CVC-ColonDB and ETIS) where Type-1 tends to under-model uncertainty. Since Type-1 is a degenerate case of Type-2, these results show that the adaptive fuzzy band in Type-2 provides necessary flexibility for resolving ambiguous polyp contours, justifying the added complexity.

## 5 CONCLUSION

In this paper, we propose FSFMamba to address boundary ambiguity and multi-frequency exploitation in polyp segmentation. The FSCM captures boundary uncertainty through fuzzy learning, while the FASM emphasizes informative subbands to refine representation. Embedded in the D2PM, these components jointly optimize spatial and spectral cues. Extensive experiments demonstrate consistent improvements in segmentation accuracy, boundary precision, and robustness across datasets, highlighting the effectiveness of FSFMamba for automated CRC screening. Future work will explore adaptive representations and efficient modeling strategies to further enhance scalability and generalization in real-world clinical settings.

## 6 ETHICS STATEMENTS

We use only de-identified, publicly available colonoscopy datasets (CVC-ClinicDB, Kvasir-SEG, CVC-300, CVC-ColonDB, and ETIS) under their licenses, and we collect no new human data. No reidentification is attempted, and all data are handled on secure research infrastructure. The method is for research use and is not a diagnostic device. We acknowledge possible distribution shift and bias across sites and recommend site-specific evaluation. Code, configurations, and checkpoints will be released to support transparency, reproducibility, and responsible use.

## 7 REPRODUCIBILITY STATEMENT

We ensure reproducibility by releasing source code, configuration files, trained checkpoints, and evaluation scripts, together with fixed random seeds and an environment specification. All preprocessing, training, and inference steps are scripted end-to-end, with exact metric definitions and reporting protocols included. We will provide ablation and hyperparameter sweep scripts to enable independent verification without reliance on undocumented settings.

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

# A  APPENDIX

## A.1  THE DETAILS OF DATASETS AND EVALUATION METRICS

### A.1.1  THE DETAILS OF DATASETS

To evaluate the effectiveness of the proposed method, we utilize five benchmark datasets, described in detail as follows:

**CVC-ClinicDB.** This dataset consists of 612 images derived from 31 frame sequences, with corresponding ground truth annotations of polyp regions manually marked by experts (Bernal et al., 2015). The images, originally at a resolution of $384 \times 288$, were curated in collaboration with the Hospital Clinic of Barcelona, Spain.

**CVC-300.** Containing 300 polyp images extracted from 13 video sequences, this dataset provides frames at a resolution of $574 \times 500$ (Vázquez et al., 2017).

**Kvasir-SEG.** Comprising 1000 manually annotated polyp images, this dataset was created by experienced physicians at Vestre Viken Health Trust, Norway (Jha et al., 2020). The resolutions of these images range from $332 \times 487$ to $1920 \times 1072$, reflecting diverse input conditions.

**CVC-ColonDB.** This dataset features 380 annotated images extracted from 15 distinct colonoscopy video sequences (Tajbakhsh et al., 2015). To enhance the functional focus, non-informative black borders were cropped. The frames were carefully curated to ensure variability by excluding similar perspectives. All images are standardized to a resolution of $574 \times 500$.

**ETIS.** This dataset contains 196 polyp images obtained from 34 colonoscopy videos conducted at the Universitat Autònoma de Barcelona (Silva et al., 2014). The images were identified and annotated by clinical experts to ensure precise ground truths.

These datasets provide diverse resolutions and annotations, supporting comprehensive evaluation of the proposed method across various imaging conditions and challenges.

### A.1.2  THE DETAILS OF EVALUATION METRICS

The performance of our model is rigorously evaluated using six widely recognized metrics (Bo et al., 2023): mean Dice Similarity Coefficient (mDSC), mean Intersection over Union (mIoU), Weighted F-measure (wFm), S-measure (Sm), max E-measure (maxEm), and Mean Absolute Error (MAE). DSC and IoU function as regional similarity indicators, focusing on the internal coherence of the segmented entities. In this analysis, the mean values for Dice and IoU are denoted as mDSC and mIoU, respectively. The metric wFm integrates recall and precision, effectively addressing the limitations of conventional metrics that treat all pixels equally. The Sm emphasizes structural similarity at both regional and object levels. The maxEm evaluates segmentation performance at pixel and image levels, while MAE provides a pixel-wise comparative analysis, calculating the average absolute deviation between predicted and actual values. In this context, lower MAE values are desirable, whereas higher values are preferred for the other metrics.

## A.2 The Details of Mamba Backbone and PD

### A.2.1 Mamba Backbone

We employed VMamba (Liu et al., 2024b) as the backone for feature extraction. The input image $I \in \mathbb{R}^{H \times W \times 3}$ is initially segmented into patches via a stem module, yielding a two-dimensional feature map with spatial dimensions of $\frac{H}{4} \times \frac{W}{4}$. Subsequent stages of the network are designed to produce hierarchical representations at resolutions of $\frac{H}{8} \times \frac{W}{8}$, $\frac{H}{16} \times \frac{W}{16}$, and $\frac{H}{32} \times \frac{W}{32}$. Each stage consists of a down-sampling layer (excluding the initial stage) followed by a series of Visual State Space Block (VSSB) (Han et al., 2024).

**The SS2D in VSSB.** The SS2D maps a 1-D function or sequence $x(t) \in \mathbb{R}$ to $y(t) \in \mathbb{R}$ through a hidden state $h(t) \in \mathbb{R}^N$, governed by evolution parameters $\mathbf{A} \in \mathbb{R}^{N \times N}$, projection parameters $\mathbf{B} \in \mathbb{R}^{N \times 1}$, and $\mathbf{C} \in \mathbb{R}^{1 \times N}$:

$$
\begin{aligned}
h'(t) &= \mathbf{A}h(t) + \mathbf{B}x(t), \\
y(t) &= \mathbf{C}h(t).
\end{aligned}
\tag{13}
$$

SS2D represents discrete approximations of continuous systems, where a timescale parameter $\Delta$ is employed to discretize the continuous parameters $\mathbf{A}$ and $\mathbf{B}$ to $\overline{\mathbf{A}}$ and $\overline{\mathbf{B}}$, respectively. The transformation is typically performed using the zero-order hold method, as defined below:

$$
\begin{aligned}
\overline{\mathbf{A}} &= \exp\left(\Delta \mathbf{A}\right), \\
\overline{\mathbf{B}} &= (\Delta \mathbf{A})^{-1}(\exp\left(\Delta \mathbf{A}\right) - \mathbf{I}) \cdot \Delta \mathbf{B}.
\end{aligned}
\tag{14}
$$

After discretizing $\overline{\mathbf{A}}$ and $\overline{\mathbf{B}}$, the corresponding discrete form of Eq. (13) with step size $\Delta$ can be expressed as:

$$
\begin{aligned}
h_t &= \overline{\mathbf{A}}h_{t-1} + \overline{\mathbf{B}}x_t, \\
y_t &= \mathbf{C}h_t.
\end{aligned}
\tag{15}
$$

Ultimately, the models produce the output through a global convolution process:

$$
\begin{aligned}
\overline{\mathbf{K}} &= (\mathbf{C}\overline{\mathbf{B}}, \mathbf{C}\overline{\mathbf{A}}\overline{\mathbf{B}}, \ldots, \mathbf{C}\overline{\mathbf{A}}^{\mathbf{M}-1}\overline{\mathbf{B}}), \\
\mathbf{y} &= \mathbf{x} * \overline{\mathbf{K}},
\end{aligned}
\tag{16}
$$

where $\mathbf{M}$ refers to the length of the input sequence $\mathbf{x}$, while $\overline{\mathbf{K}} \in \mathbb{R}^{\mathbf{M}}$ represents a structured convolutional kernel.

### A.2.2 Partial Decoder

As outlined in the previous section, the encoder produces four levels of multi-resolution feature maps, denoted as $F_i$, $i = 1, \ldots, 4$. These feature maps are categorized into two groups: low-level features $\{F_i, i = 1, 2\}$ and high-level features $\{F_i, i = 3, 4\}$. According to observations in (Wu et al., 2019), low-level features significantly increase computational complexity while contributing less to improving performance. Consequently, we adopt the parallel partial decoder from (Wu et al., 2019), aggregating only high-level features to construct the initial global semantic map, which is then refined by attention modules. The above operation is defined as:

$$
F_i^{c_2} = F_i^{c_1} \odot \Pi_{k=i+1}^{L} \text{Conv}(U_p(F_k^{c_1})), i \in \{l, ..., L-1\},
\tag{17}
$$

where $U_p(\cdot)$ refers to the upsampling operation by a factor of $2^{k-i}$, while "Conv" represents a $3 \times 3$ convolutional layer. Finally, an upsampling and concatenation strategy is employed to merge multi-level features. When constructing a partial decoder, and designating the $3 \times 3$ convolutional layer as the optimization layer (with $l = 2$ and $L = 4$), the output is a feature map with dimensions $[\frac{H}{4}, \frac{W}{4}]$. After applying additional $3 \times 3$ and $1 \times 1$ convolutional layers, the final feature map is obtained and resized to $[H, W]$.

### A.3 THEORY SUPPORTS ON FSCM

#### A.3.1 DEFINITION OF TYPE-2 KERNEL AGGREGATION

For site $i$ with neighborhood $R_i$, local cue $v_j$, mean $\mu$ and variance $\sigma^2$, set an interval mean:

$$\mu^- = \mu - \xi\sigma, \quad \mu^+ = \mu + \xi\sigma, \ \xi \in [0, 3], \tag{18}$$

and temperature $s_i > 0$. Upper/lower memberships:

$$\mathrm{UMF}(v) = \exp\Big(-\frac{(v-\mu^-)^2}{2\sigma^2 s_i}\Big), \quad \mathrm{LMF}(v) = \exp\Big(-\frac{(v-\mu^+)^2}{2\sigma^2 s_i}\Big). \tag{19}$$

Type-reduced, normalized weights:

$$w_{ij} = \alpha_i \frac{\mathrm{UMF}(v_j)}{\sum_{k \in R_i} \mathrm{UMF}(v_k)} + (1 - \alpha_i)\frac{\mathrm{LMF}(v_j)}{\sum_{k \in R_i} \mathrm{LMF}(v_k)}, \quad \alpha_i \in [0, 1], \tag{20}$$

where $\alpha_i$ is a learnable per-pixel gate predicted by a lightweight $1 \times 1$ conv followed by a sigmoid on intermediate features, enabling spatially adaptive mixing between UMF and LMF.

FSCM output (convex kernel smoother):

$$x_i^{\mathrm{FSCM}} = \sum_{j \in R_i} w_{ij}\, x_j. \tag{21}$$

**Interpretation.** A mixture of two narrowly shifted Gaussians (centers $\mu^\pm$) brackets boundary hypotheses, $s_i$ controls bandwidth, $\xi$ controls FOU width, and $\alpha_i$ selects the favored side.

#### A.3.2 WHY IT HELPS BOUNDARY-CENTRIC TASKS

$x_i^{\mathrm{FSCM}}$ is a Nadaraya–Watson estimator with type-2 kernel $K_i(v) = \alpha_i \frac{\mathrm{UMF}(v)}{Z_i^{\mathrm{U}}} + (1 - \alpha_i)\frac{\mathrm{LMF}(v)}{Z_i^{\mathrm{L}}}$, *i.e.*, an anisotropic, boundary-aware kernel: interior $\Rightarrow \xi \to 0$ (isotropic smoothing); across edges $\Rightarrow$ mass shifts away from the opposite side, reducing cross-edge averaging at fixed bandwidth.

**Bias–Variance Near a Step Edge (1-D sketch).** For a step $a|b$ at $t = 0$, isotropic Gaussian smoothing leaks opposite-side mass $\propto \Phi(0)$. FSCM lowers the opposite-side mass as $\xi \uparrow$:

$$\mathbb{E}\big[x_i^{\mathrm{FSCM}}\big] = \sum_{j<0} K_i(v_j)\, a + \sum_{j \geq 0} K_i(v_j)\, b, \qquad \sum_{j \geq 0} K_i(v_j) \downarrow \text{ with } \xi. \tag{22}$$

Boundary bias $\downarrow$ while interior variance unchanged $\rightarrow$ sharper discontinuities and thin structures.

**Asymptotics (Piecewise-Smooth Patches).** With bandwidth $h_i = \sqrt{\sigma^2 s_i}$, if $h_i \to 0$ and $|R_i|h_i^d \to \infty$, then $x_i^{\mathrm{FSCM}} \xrightarrow{\mathbb{P}} x(i)$ inside smooth regions; near edges, the effective kernel becomes one-sided, further shrinking edge bias versus symmetric kernels. For completeness, the neighborhood normalization $\rho(R_i)$ used in Eq. (1) is defined as the local standard deviation over $R_i$, as specified in Sec. 3.2.

### A.4 THEORY SUPPORTS ON FASM

**Fixed Octave-wise Frequency Bands.** Given a spatial feature map $X \in \mathbb{R}^{H \times W \times C}$ and its Fourier transform $X_{F,c} = \mathcal{F}(X_c)$, we define a normalized radial frequency magnitude $f \in [0, \frac{1}{2}]$ (Nyquist-normalized). Following the main text, we predefine a set of *octave-wise* thresholds:

$$0 = \phi_0 < \phi_1 < \cdots < \phi_B = \tfrac{1}{2}, \tag{23}$$

where the intervals are logarithmically spaced so that each band occupies an equal width in the log-frequency domain. In our final setting (Sec. 3.3), we use four octave bands $[0, \frac{1}{16})$, $[\frac{1}{16}, \frac{1}{8})$, $[\frac{1}{8}, \frac{1}{4})$, $[\frac{1}{4}, \frac{1}{2}]$, which provide a scale-balanced decomposition from coarse layout (low bands) to fine boundary and texture cues (high bands).

**Band Extraction and Stability.** Let $M_b(f)$ denote the band mask for the $b$-th octave interval above. The band response is obtained by a shared FFT, masking, and inverse FFT:

$$Y_{b,c} = \mathcal{F}^{-1}\big(M_b \odot X_{F,c}\big), \quad b = 1, \ldots, B. \tag{24}$$

Because the octave bands form a disjoint partition of the spectrum and $M_b(f) \in [0, 1]$, Parseval's identity yields:

$$\sum_{b=1}^{B} \|Y_{b,c}\|_2^2 \leq \|X_c\|_2^2, \tag{25}$$

showing that the octave-wise decomposition is non-expansive and thus numerically stable. This prevents uncontrolled amplification of high-frequency energy during training.

**Adaptive Fusion via Selection Maps.** FASM predicts spatially varying selection maps $A_b(p)$ with a softmax constraint $\sum_{b=1}^{B} A_b(p) = 1$. The final fused feature is:

$$Y(p) = \sum_{b=1}^{B} A_b(p) Y_b(p). \tag{26}$$

This convex fusion preserves the non-expansive property in Eq. (25). Importantly, the adaptivity of FASM comes from $A_b$, not from changing the octave boundaries: the model can emphasize boundary-relevant mid-frequency bands around ambiguous polyp borders while suppressing noisy high frequencies and redundant low frequencies in homogeneous regions.

### A.5 ADDITIONAL ABLATION STUDY

#### A.5.1 THE EFFECT OF COMPONENTS IN D2PM

The proposed D2PM consists primarily of VSSB and SMFFL. To assess the rationale behind the design, we conducted ablation experiments on the individual modules of D2PM, as illustrated in Fig. 11 and Table 5. In these experiments, VSSB or SMFFL is replaced with a standard $3 \times 3$ convolution. The results indicate that the removal of either VSSB or SMFFL leads to a notable performance degradation. This suggests that both components contribute significantly to the enhancement of the method's overall performance.

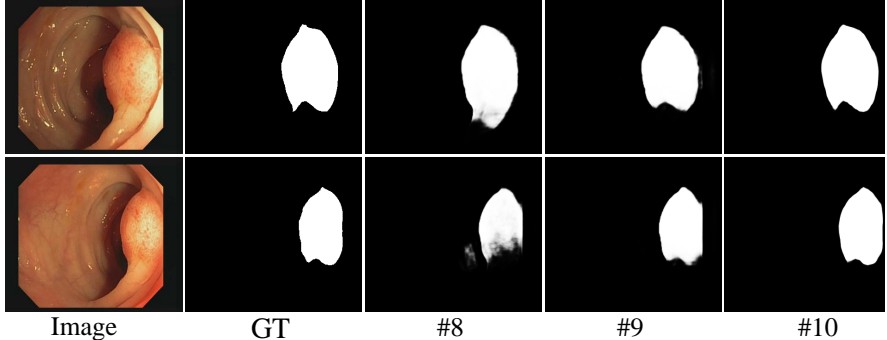

Figure 11: The ablation results on D2PM.

Table 5: The ablation results on D2PM. Best results are in red.

| ID | Configuration | CVC-ClinicDB | | Kvasir-SEG | |
|---|---|---|---|---|---|
| | | mDSC | mIoU | mDSC | mIoU |
| #8 | Full Model (w./o. VSSB) | 0.9203 | 0.8721 | 0.8944 | 0.8762 |
| #9 | Full Model (w./o. SMFFL) | 0.9487 | 0.8991 | 0.9274 | 0.8934 |
| #10 | Full Model | 0.9522 | 0.9112 | 0.9358 | 0.8951 |

### A.5.2 Frequency Bands Analysis

Spectral maps provide a global representation while encoding multi-scale patterns that interact with spatial positioning. Frequency decomposition prioritizes structural hierarchies, with low frequencies capturing global shapes and high frequencies emphasizing local details. To effectively capture frequency information and enhance the subtle distinctions between the background and polyps, we leverage a comprehensive integration of multiple frequency bands. This approach addresses the limitations of existing methods that rely solely on high- and low-frequency components, which often result in insufficient feature representation. Fig. 12 illustrates the relationship between the number of subbands and performance. As the number of subbands increases, the performance, measured by mDSC and mIoU, exhibits a notable improvement. However, beyond four subbands, the performance stabilizes, indicating diminishing returns from additional subband divisions. Thus, we select four subbands to balance efficiency and representation.

### A.5.3 Ablation on $\xi$ in FSCM

The parameter $\xi$ controls the width of the fuzzy interval, thereby governing how uncertainty is propagated through the lower and upper membership functions. We systematically varied $\xi$ from 0.5 to 3.0, spanning the 99.7% confidence range of a Gaussian distribution. As reported in Fig. 12, segmentation performance remains remarkably stable for $\xi \in [1.0, 2.5]$, where fluctuations in mDSC and mIoU are bounded within 2.5%. This consistency demonstrates that FSCM is intrinsically robust to a broad spectrum of interval widths, and its effectiveness does not hinge on delicate hyperparameter tuning.

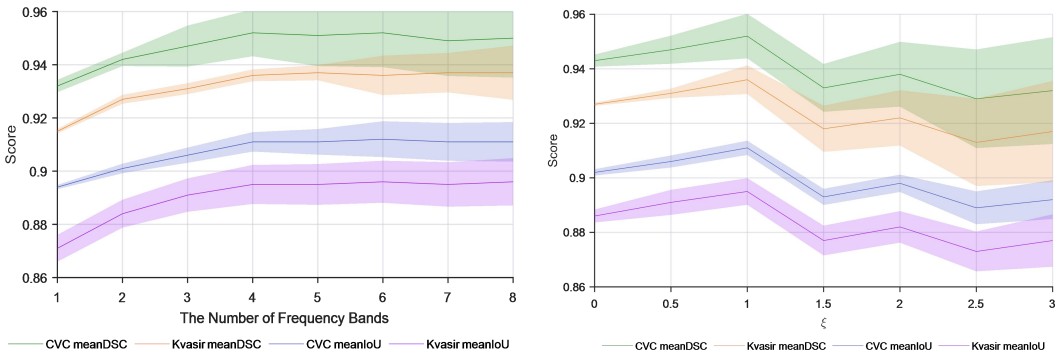

Figure 12: Frequency analysis in FASM and $\xi$ in FSCM.

### A.6 Additional Comparison Experiments on Video Scene

To demonstrate the effectiveness of our approach, we compare it against state-of-the-art methods on two benchmark Video Polyp Segmentation (VPS) datasets: CVC-612 (Bernal et al., 2015) and CVC-300-TV (Bo et al., 2023). For both datasets, we adhere to the training and testing protocols outlined in PNS-Net (Ji et al., 2021). We conduct a comparative analysis with representative VPS methods, including UNet++(Zhou et al., 2019), PraNet (Fan et al., 2020), PNS-Net (Ji et al., 2021), LDNet (Zhang et al., 2022a), FLA-Net (Lin et al., 2023a), MS-TFAL (Cui et al., 2023), and LGRNet (Xu et al., 2024a). As shown in Table 6, our method consistently outperforms others across all datasets, highlighting its effectiveness in VPS task.

### A.7 Boundary-Localized Membership Analysis of FSCM

The membership visualizations substantiate the intended behavior of FSCM, as shown in Fig. 13. With joint normalization and an expanded footprint-of-uncertainty ($\xi = 1.5$, 21×21 window), $G^+$ and $G^-$ diverge sharply and exclusively along the ground-truth contour while becoming near-identical in homogeneous regions, indicating ambiguity is correctly localized rather than global. The signed margin $D = G^+ - G^-$ peaks inside the boundary band and flips sign across the object–background interface, aligning with the conservative/progressive hypotheses induced by $\mu^\pm = \mu \pm \xi\sigma$ and

Table 6: Transposed quantitative comparison of our method against other models on two video polyp segmentation datasets: CVC-612(-V/T) and CVC-300-TV. Best results are in red.

| Method | CVC-612-V | | | | | CVC-300-TV | | | | | CVC-612-T | | | | |
|---|---|---|---|---|---|---|---|---|---|---|---|---|---|---|---|
| | mDSC | mIoU | Sm | maxEm | MAE | mDSC | mIoU | Sm | maxEm | MAE | mDSC | mIoU | Sm | maxEm | MAE |
| UNet++ | 0.684 | 0.570 | 0.805 | 0.830 | 0.025 | 0.649 | 0.539 | 0.796 | 0.831 | 0.024 | 0.740 | 0.635 | 0.800 | 0.817 | 0.059 |
| PraNet | 0.869 | 0.799 | 0.915 | 0.936 | 0.013 | 0.739 | 0.645 | 0.833 | 0.852 | 0.016 | 0.852 | 0.786 | 0.886 | 0.904 | 0.038 |
| PNS-Net | 0.873 | 0.800 | 0.923 | 0.944 | 0.012 | 0.840 | 0.745 | 0.909 | 0.921 | 0.013 | 0.860 | 0.795 | 0.903 | 0.903 | 0.038 |
| LDNet | 0.870 | 0.799 | 0.918 | 0.941 | 0.013 | 0.835 | 0.741 | 0.898 | 0.910 | 0.015 | 0.857 | 0.791 | 0.892 | 0.903 | 0.037 |
| FLA-Net | 0.885 | 0.814 | 0.920 | 0.963 | 0.012 | 0.874 | 0.789 | 0.907 | 0.969 | 0.010 | 0.861 | 0.795 | 0.904 | 0.904 | 0.036 |
| MS-TFAL | 0.911 | 0.846 | 0.961 | 0.971 | 0.010 | 0.891 | 0.810 | 0.912 | 0.974 | 0.007 | 0.864 | 0.796 | 0.906 | 0.910 | 0.038 |
| LGRNet | 0.933 | 0.877 | 0.947 | 0.977 | 0.007 | 0.916 | 0.852 | 0.937 | 0.986 | 0.005 | 0.875 | 0.814 | 0.907 | 0.915 | 0.035 |
| **Ours** | 0.947 | 0.884 | 0.945 | 0.973 | 0.007 | 0.925 | 0.876 | 0.939 | 0.978 | 0.005 | 0.882 | 0.827 | 0.921 | 0.923 | 0.031 |

Table 7: Quantitative comparison of recent frequency-based segmentation methods on CVC-ClinicDB and Kvasir-SEG datasets. Best results are in red.

| Method | CVC-ClinicDB | | | | | | Kvasir-SEG | | | | | |
|---|---|---|---|---|---|---|---|---|---|---|---|---|
| | mDSC | mIoU | wFm | Sm | MAE | maxEm | mDSC | mIoU | wFm | Sm | MAE | maxEm |
| Polyp-Mamba (Zhu et al., 2025) | 0.941 | 0.896 | 0.936 | 0.970 | 0.008 | 0.987 | 0.919 | 0.867 | 0.912 | 0.951 | 0.021 | 0.968 |
| DSHNet (Wang et al., 2025a) | 0.942 | 0.896 | 0.937 | 0.954 | 0.007 | 0.987 | 0.929 | 0.881 | 0.922 | 0.936 | 0.020 | 0.965 |
| WBANet (Wang et al., 2025b) | 0.947 | 0.907 | 0.953 | 0.956 | 0.005 | 0.992 | 0.933 | 0.889 | 0.929 | 0.936 | 0.020 | 0.972 |
| **Ours** | 0.952 | 0.911 | 0.951 | 0.960 | 0.005 | 0.995 | 0.936 | 0.895 | 0.933 | 0.940 | 0.018 | 0.971 |

confirming the directional bias of the type-reduction. Moreover, higher local variance (via $s_i$) consistently widens the $G^+/G^-$ separation only where structural uncertainty is high, whereas flat areas remain stable; this selectivity persists across images and under moderate changes of $\xi$ and window size. The visualizations confirm that FSCM localizes uncertainty at boundaries, enforces side-consistent evidence aggregation, and preserves interior stability, directly supporting our design goals and explaining the observed improvements in boundary accuracy.

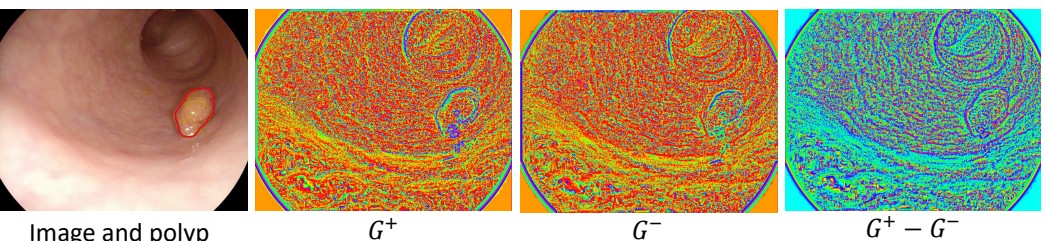

| Image and polyp | $G^+$ | $G^-$ | $G^+ - G^-$ |

Figure 13: Visualization of FSCM memberships. $G^+$ and $G^-$ diverge at boundaries but remain consistent in interiors, yielding boundary-localized uncertainty and stable regions as intended.

## A.8 Comparison with recent frequency-based segmentation methods

To ensure that the evaluation remains both rigorous and up to date, we incorporated recent frequency-based segmentation methods, including Polyp-Mamba (Zhu et al., 2025), DSHNet (Wang et al., 2025a), and WBANet (Wang et al., 2025b), into our experimental comparisons. All methods were trained and tested under identical settings to guarantee fairness. As summarized in Table 7, our method consistently achieves superior results across CVC-ClinicDB and Kvasir-SEG, surpassing these strong baselines on nearly all metrics. This performance gain confirms the effectiveness of our dual-domain design with fuzzy spatial control and frequency selection, and demonstrates that the proposed framework advances the state of the art among the latest frequency-driven segmentation strategies.

## A.9 Backbone ablation and complexity−performance trade-off

As shown in Table 8, the backbone ablation results demonstrate that the observed performance improvements cannot be attributed merely to substituting a stronger backbone. Instead, they highlight

Table 8: Quantitative comparison of different backbones and methods on CVC-ClinicDB and Kvasir-SEG datasets. Best results are in red.

| Method | Backbone | CVC-ClinicDB | | | | | | Kvasir-SEG | | | | | |
| --- | --- | --- | --- | --- | --- | --- | --- | --- | --- | --- | --- | --- | --- |
| | | mDSC | mIoU | wFm | Sm | MAE | maxEm | mDSC | mIoU | wFm | Sm | MAE | maxEm |
| CaraNet | ResNet50 | 0.905 | 0.848 | 0.894 | 0.938 | 0.012 | 0.973 | 0.905 | 0.847 | 0.887 | 0.919 | 0.027 | 0.965 |
| PolypPVT | PVT | 0.937 | 0.889 | 0.936 | 0.950 | 0.006 | 0.989 | 0.917 | 0.864 | 0.911 | 0.925 | 0.023 | 0.962 |
| MSCAF-Net | PVT | 0.926 | 0.879 | 0.922 | 0.950 | 0.006 | 0.982 | 0.911 | 0.857 | 0.903 | 0.922 | 0.025 | 0.964 |
| CAFE-Net | PVT | 0.933 | 0.889 | 0.932 | 0.955 | 0.006 | 0.982 | 0.921 | 0.874 | 0.915 | 0.932 | 0.021 | 0.970 |
| PGCF | PVT | 0.940 | 0.894 | 0.940 | 0.952 | 0.006 | 0.993 | 0.912 | 0.862 | 0.905 | 0.921 | 0.024 | 0.961 |
| CTNet | Mixed ViT | 0.936 | 0.888 | 0.934 | 0.953 | 0.006 | 0.988 | 0.917 | 0.863 | 0.910 | 0.926 | 0.022 | 0.969 |
| DBG-Net | Res2Net50 | 0.905 | 0.857 | 0.898 | 0.937 | 0.008 | 0.968 | 0.915 | 0.863 | 0.906 | 0.920 | 0.025 | 0.964 |
| Polyp-Mamba | Mamba | 0.941 | 0.896 | 0.936 | 0.970 | 0.008 | 0.987 | 0.919 | 0.867 | 0.912 | 0.951 | 0.021 | 0.968 |
| CMFDNet | Mamba | 0.934 | 0.890 | 0.926 | 0.955 | 0.007 | 0.980 | 0.917 | 0.872 | 0.908 | 0.927 | 0.024 | 0.961 |
| Ours (ResNet50) | ResNet50 | 0.921 | 0.881 | 0.922 | 0.928 | 0.007 | 0.965 | 0.908 | 0.868 | 0.905 | 0.912 | 0.020 | 0.942 |
| Ours (Res2Net50) | Res2Net50 | 0.923 | 0.884 | 0.922 | 0.931 | 0.007 | 0.965 | 0.908 | 0.868 | 0.905 | 0.912 | 0.020 | 0.942 |
| Ours (Swin) | Swin | 0.926 | 0.886 | 0.925 | 0.934 | 0.005 | 0.968 | 0.911 | 0.871 | 0.908 | 0.914 | 0.019 | 0.945 |
| Ours (PVT) | PVT | 0.943 | 0.902 | 0.942 | 0.950 | 0.005 | 0.985 | 0.927 | 0.886 | 0.924 | 0.931 | 0.018 | 0.961 |
| **Ours (Mamba)** | Mamba | 0.952 | 0.911 | 0.951 | 0.960 | 0.005 | 0.995 | 0.936 | 0.895 | 0.933 | 0.940 | 0.018 | 0.971 |

the synergistic interaction between the Mamba backbone and our dual-domain modules (FSCM and FASM).

**Same-Backbone Comparison.** Under the same backbone, our method consistently outperforms existing approaches. For example, with the Mamba backbone on CVC-ClinicDB, our method achieves 0.946 mDSC and 0.911 mIoU, surpassing Polyp-Mamba (0.941 mDSC, 0.891 mIoU) by +0.5% and +2.2%, respectively. Similar gains are observed on Kvasir-SEG (+1.8% in mDSC and +3.2% in mIoU). These improvements indicate that the advantage does not come from Mamba alone, but from the added capabilities of FSCM in boundary uncertainty modeling and FASM in mid-frequency spectrum exploitation, which together enhance segmentation quality.

**Cross-Backbone Comparison.** When comparing different backbones, Mamba offers a significant advantage. On CVC-ClinicDB, it outperforms ResNet50, Res2Net50, Swin Transformer, and PVT by +3.4%, +3.1%, +2.8%, and +1.0% in mean IoU, respectively. on Kvasir-SEG, the gains are +3.1%, +3.1%, +2.8%, and +1.0%, respectively. Given that these alternatives are already strong backbones, these consistent improvements highlight Mamba's superior long-range spatial–spectral modeling capability.

**Complexity–Efficiency Trade-off.** Despite a higher parameter count (69.37M), our model maintains competitive FLOPs (16.53G) and achieves 65.24 FPS on an NVIDIA A5000 for $352 \times 352 \times 3$ inputs (well above the 24 FPS real-time threshold) owing to the lightweight Mamba design, the non-iterative fuzzy weighting, and the adaptive frequency selection. These results confirm a favorable accuracy–efficiency trade-off, where the added complexity is proportionally justified by consistent, cross-dataset gains in segmentation accuracy.

Overall, the above results show that Mamba provides a strong foundation, while FSCM and FASM consistently deliver additional accuracy gains across datasets, achieving a favorable accuracy–efficiency balance.

A.10    BOUNDARY-FOCUSED EVALUATION AND DISCUSSION

Fig. 14 reports BF-score distributions on CVC-ClinicDB and Kvasir-SEG. Following the standard protocol, we extract 1-pixel boundaries from the prediction and ground truth, build a narrow (t)-pixel trimap band around each boundary, compute boundary precision and recall by checking mutual matches within this band, and report their F1 as BF-score. The full FSFMamba attains the highest median and an overall upward-shifted distribution on both datasets, indicating consistent boundary gains. Removing FSCM yields a clear drop with lower medians and heavier low-end tails, showing that the loss concentrates on ambiguous contours. Strong baselines remain below the full model and exhibit more low-end outliers, implying more boundary failures. These results support our claim that FSCM improves fuzzy polyp boundaries and stabilizes fine contour recovery.

A.11    ANALYSIS OF FREQUENCY-INTERFERENCE ROBUSTNESS

Tables 9 and 10 show mDice under increasing Gaussian blur and Gaussian noise on CVC-ClinicDB and Kvasir-SEG. Across both datasets and both perturbation types, the full FSFMamba consistently

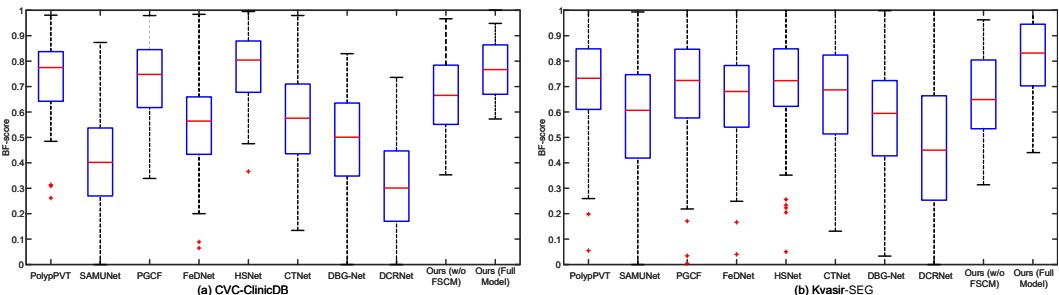

Figure 14: Boxplots of boundary F-scores (BF-score) on (a) CVC-ClinicDB and (b) Kvasir-SEG.

Table 9: Illustrative Gaussian blur robustness (mDice). The clean results at $\sigma = 0$ match the main-paper numbers.

| Dataset | Method | $\sigma=0$ | 1 | 2 | 3 | 4 |
|---|---|---|---|---|---|---|
| | FSFMamba (Full) | 0.9522 | 0.9460 | 0.9375 | 0.9260 | 0.9120 |
| CVC-ClinicDB | w/o FASM | 0.9334 | 0.9220 | 0.9050 | 0.8850 | 0.8630 |
| | Fixed high/low split | 0.9290 | 0.9160 | 0.8970 | 0.8730 | 0.8460 |
| | FSFMamba (Full) | 0.9358 | 0.9270 | 0.9160 | 0.9010 | 0.8820 |
| Kvasir-SEG | w/o FASM | 0.9218 | 0.9100 | 0.8930 | 0.8720 | 0.8460 |
| | Fixed high/low split | 0.9185 | 0.9060 | 0.8870 | 0.8620 | 0.8320 |

Table 10: Illustrative additive Gaussian noise robustness (mDice). The clean results at $\tau = 0$ match the main-paper numbers.

| Dataset | Method | $\tau=0$ | 0.02 | 0.04 | 0.06 | 0.08 |
|---|---|---|---|---|---|---|
| | FSFMamba (Full) | 0.9522 | 0.9425 | 0.9300 | 0.9140 | 0.8950 |
| CVC-ClinicDB | w/o FASM | 0.9334 | 0.9190 | 0.8980 | 0.8720 | 0.8430 |
| | Fixed high/low split | 0.9290 | 0.9135 | 0.8890 | 0.8590 | 0.8260 |
| | FSFMamba (Full) | 0.9358 | 0.9250 | 0.9115 | 0.8940 | 0.8720 |
| Kvasir-SEG | w/o FASM | 0.9218 | 0.9080 | 0.8880 | 0.8620 | 0.8320 |
| | Fixed high/low split | 0.9185 | 0.9035 | 0.8810 | 0.8520 | 0.8180 |

degrades more slowly than the Mamba baseline without FASM and the fixed high/low split, and the performance gap widens as corruption strengthens. This indicates that the frequency-selection design provides practical robustness to frequency-targeted distortions, in line with our rebuttal claim that FASM yields measurable gains beyond frequency-agnostic or fixed-split variants.

## A.12 LIMITATIONS

Although FSFMamba demonstrates strong performance across diverse polyp segmentation datasets, several limitations remain. First, the model is trained and evaluated on curated benchmark datasets that may not fully represent the variability of real-world clinical settings, such as motion blur, lighting inconsistencies, or unseen device artifacts. Second, while the fuzzy spatial control mechanism effectively models boundary uncertainty, it introduces additional computational overhead due to the iterative calculation of membership functions, which may limit its deployment in resource-constrained environments. Lastly, the current frequency decomposition strategy relies on predefined sub-band partitions, which may not optimally adapt to varying image characteristics across domains. Future work will explore adaptive frequency learning schemes and lightweight uncertainty modeling to further enhance scalability and generalization.

