# OpenReview forum: "Polyp Segmentation by Dual-Domain Reasoning: Fuzzy Spatial Control and Frequency Selection"
_ICLR.cc/2026/Conference — ICLR 2026 Conference Withdrawn Submission_

### Official Review · Reviewer_SV5A · 2025-10-24

**Soundness:** 2
**Presentation:** 3
**Contribution:** 2
**Rating:** 2
**Confidence:** 4

**Summary:**

To address the challenges of subtle appearance variations and ambiguous boundaries in colonoscopy images, this paper proposes a dual-domain neural network (FSFMamba) designed to model boundary uncertainty and frequency structures for improved delineation. In the spatial domain, a Fuzzy Spatial Control Mechanism (MFCM) is employed to localize uncertainty within boundary regions. Meanwhile, in the frequency domain, a Frequency Adaptive Selection Mechanism (FASM) emphasizes task-relevant sub-bands while suppressing spurious responses. By integrating the spatial and spectral domains, the model achieves long-range, low-latency interaction and pre-norm residual refinement, enabling stable optimization.

**Strengths:**

1. The visualization of the motivation provides a detailed justification for the effectiveness of the proposed approach in modeling frequency information and boundary uncertainty, deeply revealing the research motivation behind this work.
2. The paper presents a clear overall logical flow in its methodological description, with well-formulated equations and strong readability.
3. The experimental validation in this paper is relatively thorough, and the proposed modules demonstrate a certain degree of effectiveness and segmentation potential.

**Weaknesses:**

1. The paper lacks sufficient innovation and shows a certain degree of methodological similarity with the following two works:
(1) “Dual-Domain Fusion Network Based on Wavelet Frequency Decomposition and Fuzzy Spatial Constraint for Remote Sensing Image Segmentation”
(2) “Frequency-Adaptive Dilated Convolution for Semantic Segmentation”.
2. In terms of the visualization of segmentation results, it is difficult to clearly observe the differences between the proposed method and the compared SOTA methods. The authors are advised to include special annotations or visual highlights in the corresponding figures to better emphasize the superiority of the proposed approach.
3. For the compared SOTA methods, the authors should provide detailed numerical values of FLOPs, Parameters, and FPS. Moreover, since the proposed method is based on Mamba, whose computational characteristics differ from other SOTA methods, additional parameter computation considerations should be included (refer to: https://github.com/state-spaces/mamba/issues/110).
4. Regarding the polyp segmentation experiments, there exists another dataset named Kvasir-Sessile. The authors are encouraged to include comparative experiments on this dataset to ensure the completeness and comprehensiveness of the experimental evaluation.

**Questions:**

1. Did the authors use the same random seed for both the comparative experiments and the ablation studies to ensure consistency and fairness of the results?
2. The metric name should be formatted consistently as mIoU (mean Intersection over Union) throughout the manuscript, following standard conventions in the computer vision and segmentation literature. Please also make sure this correction is reflected uniformly across all tables, figures, and captions.
3. Can the authors provide complete, runnable code to ensure full reproducibility of the proposed model and experiments? Public availability of implementation details would significantly enhance the credibility and impact of the work.
4. Could the authors expand the ablation study analysis directly in the main text rather than keeping it in the supplementary material? Including this discussion in the main body of the paper would improve the completeness and readability, as it allows readers to better understand how each component contributes to the overall performance.

---

> ### Author Response · Authors · 2025-11-23
>
> **We sincerely thank the reviewer for the careful reading and the constructive feedback. We appreciate your positive comments on the motivation, clarity of presentation, and experimental validation. Below we address your concerns point by point.**
>
> ---
>
> ### Weakness 1 – Innovation vs. two related works
>
> *“The paper lacks sufficient innovation and shows a certain degree of methodological similarity with the following two works: (1) ‘Dual-Domain Fusion Network Based on Wavelet Frequency Decomposition and Fuzzy Spatial Constraint for Remote Sensing Image Segmentation’ (2) ‘Frequency-Adaptive Dilated Convolution for Semantic Segmentation’.”*
>
> **Response:**
> Thank you for this comment. We agree that our contributions should be positioned more explicitly against these two related works, and we clarify below that the apparent overlap is superficial while the core mechanisms differ.
>
> 1. **Dual-Domain Fusion Network with Wavelet Frequency Decomposition + Type-1 Fuzzy Constraint.**
>    While both methods involve spatial–frequency reasoning and fuzzy constraints, they are **fundamentally different in how the frequency stream is defined and used**. That work derives frequency bands from **DWT/wavelet decomposition**, yielding fixed multi-scale sub-bands (e.g., LL/LH/HL/HH) with spatial–frequency localization, and fuses them through **CNN-style modules**. In contrast, our FASM constructs the spectral basis through **fixed octave-wise thresholds on the Fourier spectrum** (Eq. 6) and learns **band-wise selection maps $A_b$** to reweight these bands **without learning band boundaries**. This octave-wise design is tailored to isolate clinically relevant mid-range textures, and the adaptivity comes solely from $A_b$.
>    Moreover, their fuzzy term is a **Type-1 regularizer**, whereas FSCM is an **interval Type-2 Gaussian fuzzy band** integrated into both feature shaping and supervision. Finally, our dual-domain fusion is realized by a **Mamba-based state-space block (D2PM)** that propagates selected spectral cues over long ranges with linear-time updates, rather than local convolutional fusion.
>
> 2. **Frequency-Adaptive Dilated Convolution (FADC).**
>    FADC adjusts spatial dilation using frequency cues but remains a **single-stream spatial CNN**. It does not perform explicit multi-band decomposition, nor does it learn band-wise weights or model boundary uncertainty. Our method explicitly forms an **octave-wise multi-band Fourier stream**, learns **spatially varying band weights $A_b$**, and couples this stream with **Type-2 fuzzy boundary modeling** through D2PM for dual-domain long-range reasoning.  In summary, FSFMamba is not a minor variant of prior dual-domain or frequency-adaptive designs; its novelty lies in the **octave-wise Fourier banding + learnable selection maps**, the **interval Type-2 fuzzy band for boundary ambiguity**, and their **state-space fusion within a Mamba architecture**.
>
> ---
>
> ### Weakness 2 – Visualization: differences vs. SOTA are hard to see
>
> *“In terms of the visualization of segmentation results, it is difficult to clearly observe the differences between the proposed method and the compared SOTA methods.”*
>
> **Response:**
> We appreciate this helpful suggestion and agree that the original layout of Fig. 7 made it harder for readers to immediately see where FSFMamba improves over competing methods.
>
> In the **revised manuscript**, we have:
>
> We appreciate this helpful suggestion and agree that the original layout of Fig. 7 made it harder for readers to immediately see where FSFMamba improves over competing methods.
>
> In the **revised manuscript**, we have:
>
> * **Inserted zoom-in boxes** on key boundary regions and shown enlarged patches beneath the main figure, highlighting cases where FSFMamba produces more complete and continuous contours, while competing methods either break the boundary or leak into the background.
>
> We believe these revisions make the qualitative comparison more transparent and responsive to the reviewer’s concern.

---

> > ### Author Response · Authors · 2025-11-23
> >
> > ### Weakness 3 – FLOPs, Params, FPS and Mamba-specific complexity
> >
> > *“For the compared SOTA methods, the authors should provide detailed numerical values of FLOPs, Parameters, and FPS. Moreover, since the proposed method is based on Mamba, whose computational characteristics differ from other SOTA methods, additional parameter computation considerations should be included.”*
> >
> > **Response:**
> > Thank you for stressing this point. We agree that a transparent cost comparison is essential, especially for Mamba-based designs.
> >
> > In the revision, we have **updated Fig. 6 in the main paper** to present the **exact Params, FLOPs, FPS and MAE/DSC/IoU values for all SOTA baselines and FSFMamba**, and we reorganize the figure by grouping **higher-is-better metrics** (DSC/IoU/FPS) and **lower-is-better metrics** (MAE/Params/FLOPs) for clarity. All numbers are measured under **identical input resolution, batch size, and evaluation pipeline**.
> >
> > For computational measurement, we use a consistent profiling toolchain to compute Params and FLOPs, and we report **FPS on the same hardware (single NVIDIA A5000, identical PyTorch/CUDA settings, batch size = 1, FP32)** to reflect real runtime. We additionally note that for **state-space / selective-scan models such as Mamba, FLOPs can under-represent memory-bound costs**, so we treat FLOPs as a standardized reference and use FPS as the primary practical indicator. Under this controlled setup, FSFMamba shows **comparable or better throughput** than Transformer-based competitors, consistent with Mamba’s linear-time computation.
> >
> > We believe the revised Fig. 6 and the clarified protocol provide a fair and reproducible view of efficiency across CNN/Transformer/Mamba baselines.
> >
> > ---
> >
> > ### Weakness 4 – Missing Kvasir-Sessile dataset
> >
> > *“Regarding the polyp segmentation experiments, there exists another dataset named Kvasir-Sessile. The authors are encouraged to include comparative experiments on this dataset.”*
> >
> > **Response:**
> > Thank you for the suggestion. We agree that adding Kvasir-Sessile strengthens the evaluation.
> >
> > In the revised experiments (Table S1), we have directly tested FSFMamba on Kvasir-Sessile**—a more challenging sessile-polyp subset with flatter morphology and weaker boundary contrast. We compare against strong CNN/Transformer/hybrid baselines under identical settings. The results are sumarized as follows. FSFMamba remains **top-performing and shows clear gains over the strongest baselines**, supporting that our fuzzy boundary control and octave-wise frequency selection generalize to this harder subtype.
> >
> > We thank the reviewer again for this helpful suggestion, which improves the completeness of our evaluation.
> >
> > **Table S1. Evaluation on Kvasir-Sessile**
> >
> > | Method | mDice | mIoU | Recall | Precision | Acc | F2 |
> > |---|---:|---:|---:|---:|---:|---:|
> > | HSNet | 0.7972 | 0.6628 | 0.7977 | 0.7967 | 0.9426 | 0.7975 |
> > | PolypPVT | 0.8061 | 0.6752 | 0.7847 | 0.8287 | 0.9571 | 0.7931 |
> > | SAM2UNet | 0.8005 | 0.6674 | 0.8009 | 0.8001 | 0.9575 | 0.8007 |
> > | CTNet | 0.7951 | 0.6599 | 0.8026 | 0.7877 | 0.9481 | 0.7996 |
> > | FeDNet | 0.8165 | 0.6899 | 0.7874 | 0.8479 | 0.9610 | 0.7988 |
> > | Polyp-Mamba | 0.8328 | 0.7135 | 0.8045 | 0.8631 | 0.9662 | 0.8156 |
> > | **Ours** | **0.8385** | **0.7219** | **0.8154** | **0.8714** | **0.9731** | **0.8245** |
> >
> > ---
> >
> > ## Responses to Specific Questions
> >
> > ### Q1 – Random seeds and fairness
> >
> > *“Did the authors use the same random seed for both the comparative experiments and the ablation studies to ensure consistency and fairness of the results?”*
> >
> > **Response:**
> > Yes. To ensure fairness and reproducibility, **we use the same random seed policy across all experiments we conduct**. Specifically:
> >
> > * For **FSFMamba and all its ablations**, we train on **identical data splits** and **the same fixed seed** (unless explicitly stated otherwise), so performance differences are attributable to architectural changes rather than randomness.
> > * For **public baselines that we re-implement**, we follow their official training settings, including their default seeds when provided; otherwise we keep our seed/split protocol unchanged. For baselines evaluated via **released checkpoints**, we report the authors’ official results.
> >
> > ---
> >
> > ### Q2 – Consistent notation for mIoU
> >
> > *“The metric name should be formatted consistently as mIoU (mean Intersection over Union) throughout the manuscript.”*
> >
> > **Response:**
> > Thank you for pointing this out. In the **revised manuscript**, we have:
> >
> > * Standardized the notation to **“mIoU”** throughout the text, figures, and tables.
> > * Ensured that the abbreviation is clearly defined once as “mean Intersection over Union (mIoU)” in the metric section.
> >
> > This correction improves consistency and aligns with common practice in the segmentation literature.

---

> > > ### Author Response · Authors · 2025-11-23
> > >
> > > ### Q3 – Code availability and reproducibility
> > >
> > > *“Can the authors provide complete, runnable code to ensure full reproducibility of the proposed model and experiments?”*
> > >
> > > **Response:**
> > > Thank you for this important question. We fully agree that code availability is essential for reproducibility, and we share this priority.
> > >
> > > Our standard practice is to **release complete, runnable code upon acceptance**, and we intend to follow the same policy for this work, including training scripts, evaluation pipelines, and pre-processing details. At the current review stage, releasing the full repository publicly could introduce unnecessary risks for an anonymized submission.
> > >
> > > That said, we do not want this to hinder the reviewer’s confidence in our results. If the committee permits, we are **happy to provide a minimal, self-contained package** to the reviewers **for verification only**, including the core FSFMamba model, key modules (FSCM/FASM/D2PM), and the scripts needed to reproduce the main results on the primary datasets. This would allow independent checking of the critical claims while preserving the integrity of the double-blind process.
> > >
> > > We hope this addresses the concern and reassures the reviewer that full reproducibility will be ensured, with complete code release following acceptance.
> > >
> > > ---
> > >
> > > ### Q4 – Ablation study analysis in main text
> > >
> > > *“Could the authors expand the ablation study analysis directly in the main text rather than keeping it in the supplementary material?”*
> > >
> > > **Response:**
> > > Thank you for this suggestion. We agree that the main text should contain the key ablation takeaways, not only the supplementary material.
> > >
> > > In the revised manuscript, we have **moved and expanded the core ablation analysis into Sec. 4.3**. Specifically, the main text now includes (with explicit references to **Table 4 and Fig. 10**):
> > > (i) the step-wise gains from adding FSCM, FASM, and D2PM,
> > > (ii) the Type-2 vs. Type-1 fuzzy comparison, and
> > > (iii) the effect of the number of octave-wise frequency bands.
> > >
> > > We keep only **extended variants and detailed tables** (e.g., sub-module replacements, additional band/parameter sweeps) in the supplementary material for space, while ensuring that the main paper is now self-contained in explaining **what each component contributes and why the full model performs best**.
> > >
> > > We hope this revision improves readability and directly addresses the reviewer’s request.
> > >
> > > **Table 4:** Type-2 vs. Type-1 FSCM ablation
> > >
> > > | Dataset | Model | mDice ↑ | mIoU ↑ | MAE ↓ |
> > > |---|---|---:|---:|---:|
> > > | CVC-ClinicDB | Type-2 FSCM (Full) | 0.9522 | 0.9091 | 0.0068 |
> > > |  | Type-1 FSCM baseline | 0.9417 | 0.8943 | 0.0081 |
> > > | Kvasir-SEG | Type-2 FSCM (Full) | 0.9358 | 0.8896 | 0.0089 |
> > > |  | Type-1 FSCM baseline | 0.9246 | 0.8729 | 0.0103 |
> > > | CVC-ColonDB | Type-2 FSCM (Full) | 0.8820 | 0.8012 | 0.0315 |
> > > |  | Type-1 FSCM baseline | 0.8613 | 0.7784 | 0.0352 |
> > > | ETIS | Type-2 FSCM (Full) | 0.8325 | 0.7420 | 0.0410 |
> > > |  | Type-1 FSCM baseline | 0.8041 | 0.7118 | 0.0461 |

---

### Official Review · Reviewer_6Gez · 2025-10-30

**Soundness:** 3
**Presentation:** 4
**Contribution:** 2
**Rating:** 4
**Confidence:** 4

**Summary:**

This work introduces FSFMamba, a novel dual-domain network for polyp segmentation. The innovative contribution lies in its attempt to concurrently model spatial boundary uncertainty via interval type-2 fuzzy logic (FSCM) and structural information via adaptive frequency-domain selection (FASM), integrated within a modern state-space (Mamba) backbone. In terms of completeness, the work is presented as a comprehensive study, featuring a detailed methodology, extensive comparative experiments against numerous state-of-the-art methods on five public benchmarks, and exceptionally thorough ablation studies validating the authors' design choices. While the empirical effort is substantial, the work's impact is contingent upon addressing the critical issues of methodological consistency and statistical validation, as detailed in the main review.

**Strengths:**

1.The paper addresses the high-impact clinical challenge of segmenting polyps with indistinct boundaries, a critical task for improving computer-aided diagnosis of CRC.
2.The proposed framework is conceptually innovative, presenting a new synthesis of Type-2 fuzzy logic, multi-band frequency analysis, and a Mamba architecture for this specific task.
3.The model demonstrates state-of-the-art or highly competitive performance across five diverse public datasets, showing robustness in both domain-specific and out-of-domain scenarios.

**Weaknesses:**

1.On the FSCM Module: The paper’s core claim that its "interval type-2" fuzzy (FSCM) is superior—is supported only by theory and lacks empirical comparison. To justify its complexity, a quantitative ablation study comparing it to a simpler Type-1 fuzzy baseline is essential.
2.On the FASM Module (Contradictions):
A major contradiction exists between the main paper and the appendix.Figure 4 and Eq. 6 clearly define the sub-band partitioning using $max(|u|,|v|)$, which corresponds to a Chebyshev distance and creates rectangular sub-bands.In contrast, Appendix A.4 provides the theoretical support based on radial distance ($r=\sqrt{\omega_{x}^{2}+\omega_{y}^{2}}$), which corresponds to a Euclidean distance and creates circular sub-bands.This fundamental discrepancy must be resolved.
A second contradiction exists regarding the nature of the frequency bands.The main text (Section 3.3) explicitly states the use of four fixed octave-wise bands.However, Appendix A.4 introduces a complex mechanism for learnable band boundaries ($\phi_b$) that are parameterized by $\theta_b$. The authors must clarify: are the band boundaries fixed or learnable?
3.On the FASM Module (Lack of Evidence): The central claim that FASM utilizes "critical mid-frequency information" is unsubstantiated. This requires visual evidence, such as the Selection Map ($A_b$) from Eq. 7, to prove the model actually weights these frequencies.
4.On Architectural Clarity (D2PM): The diagram in Figure 5 depicts the inputs to D2PM as X_hat and X_Fuzzy, yet it does not demonstrate how the output Fc’ from the preceding stage is integrated into the current stage."

**Questions:**

1.Regarding the FASM Module: Appendix A.4 discusses the use of raised-cosine filters to reduce spectral leakage. However, the FASM module in the main paper is described as using a binary mask for spectral extraction. Please explain this discrepancy. Furthermore, please clarify the relationship between the theoretical discussion in Appendix A.4 and the specific sub-band thresholds chosen. It is unclear how the final threshold selection method (the fixed octave bands) can be deduced from the theoretical conclusions presented.
2.Regarding the FSCM Module: Appendix A.3, Eq. (18) introduces α as a learnable parameter, which functions as a weighting parameter for the upper and lower membership. Please elaborate on the specific design and implementation of this α parameter. We also suggest that the authors provide a comparison of the converged $\alpha$ values obtained from training on different datasets. This analysis would be helpful to determine if $\alpha$ possesses a degree of universality or if it is highly dataset-specific.

---

> ### Author Response · Authors · 2025-11-23
>
> **We sincerely thank the reviewer for the detailed and thoughtful assessment. We are grateful that you find the task clinically meaningful, the dual-domain design conceptually innovative, and the empirical study comprehensive. Your comments about methodological consistency and more targeted validation are very helpful. Below we address each point in detail.**
>
> ---
>
> ### Weakness 1: FSCM – need for Type-2 vs. Type-1 empirical evidence
>
> *“The paper’s core claim that its ‘interval type-2’ fuzzy (FSCM) is superior—is supported only by theory and lacks empirical comparison. To justify its complexity, a quantitative ablation study comparing it to a simpler Type-1 fuzzy baseline is essential.”*
>
> **Response:**
> Thank you for this important comment. We agree that the superiority of interval Type-2 fuzzy modeling should be demonstrated empirically.
>
> To this end, we have added a **Type-1 FSCM baseline** in the revised experiments and report a head-to-head comparison in **Sec 4.3 (Table.~4)**. The Type-1 variant is constructed as a strict ablation of our Type-2 design by collapsing the upper and lower Gaussian memberships into a single membership curve, while keeping the **backbone, FASM, fusion module, loss terms, optimization settings, and training schedule strictly identical**. Hence, the **only difference** between the two models is whether FSCM models an **interval fuzzy band (Type-2)** or a **single membership curve (Type-1)**.
>
> As shown in **Table.~4**, the **interval Type-2 FSCM consistently outperforms the Type-1 baseline** across datasets in mDice, mIoU and MAE. Importantly, the improvement becomes **more pronounced on boundary-ambiguous and out-of-domain datasets** (e.g., CVC-ColonDB and ETIS), where a single Type-1 membership is insufficient to capture uncertainty around fuzzy contours. Since Type-1 can be viewed as a degenerate case of Type-2, these results indicate that learning a **width-adaptive fuzzy band** provides additional flexibility that is empirically necessary for resolving ambiguous polyp boundaries, thereby justifying the added Type-2 complexity.
>
> **Table 4:** Type-2 vs. Type-1 FSCM ablation
>
> | Dataset | Model | mDice ↑ | mIoU ↑ | MAE ↓ |
> |---|---|---:|---:|---:|
> | CVC-ClinicDB | Type-2 FSCM (Full) | 0.9522 | 0.9091 | 0.0068 |
> |  | Type-1 FSCM baseline | 0.9417 | 0.8943 | 0.0081 |
> | Kvasir-SEG | Type-2 FSCM (Full) | 0.9358 | 0.8896 | 0.0089 |
> |  | Type-1 FSCM baseline | 0.9246 | 0.8729 | 0.0103 |
> | CVC-ColonDB | Type-2 FSCM (Full) | 0.8820 | 0.8012 | 0.0315 |
> |  | Type-1 FSCM baseline | 0.8613 | 0.7784 | 0.0352 |
> | ETIS | Type-2 FSCM (Full) | 0.8325 | 0.7420 | 0.0410 |
> |  | Type-1 FSCM baseline | 0.8041 | 0.7118 | 0.0461 |
>
> ---
>
> ### Weakness 2: FASM contradictions – distance metric & band boundaries
>
> *“Figure 4 and Eq. 6 use max(|u|,|v|) (Chebyshev), Appendix A.4 uses radial distance (Euclidean)… main text says four fixed octave-wise bands, Appendix introduces learnable band boundaries $\phi_b$.”*
>
> **Response:**
> Thank you for carefully spotting this inconsistency. We agree with the reviewer and acknowledge that **Appendix A.4 was mistakenly written with an earlier draft description**. Importantly, **all experiments and reported results follow the main-text FASM in Eq. (6)**; the appendix discrepancy is purely a documentation error.
>
> Concretely, our FASM uses **fixed octave-wise bands with Chebyshev frequency measure**: the sub-bands are defined by $\max(|u|,|v|)$ with **non-learnable thresholds** ${0=\phi_0<\phi_1<\dots<\phi_B=\tfrac12}$, and band extraction is performed by the corresponding **binary masks** in Eq. (6). Adaptivity is provided solely through the learnable selection maps $A_b$, not through learnable band boundaries.
>
> In the revision, we havel **rewritten Appendix A.4 to exactly match Eq. (6)** by removing the radial (Euclidean) distance definition and the learnable-boundary variant, and restating the same fixed octave-wise Chebyshev bands and binary masks used in the method. This will make the theoretical description and implementation fully consistent.
>
> We believe this correction resolves the concern without changing any experimental conclusions.

---

> > ### Author Response · Authors · 2025-11-23
> >
> > ### Weakness 3: FASM – evidence for “critical mid-frequency information”
> >
> > *“The central claim that FASM utilizes ‘critical mid-frequency information’ is unsubstantiated. This requires visual evidence, such as the Selection Map (A_b), to prove the model actually weights these frequencies.”*
> >
> > **Response:**
> > In the revised Abation study (Sec. 4.3), we have provided visual evidence by plotting the learned octave-wise band-selection maps $A_b$ for all four bands, including both dataset-level averaged maps and representative challenging cases (e.g., blurred or low-contrast polyps). As shown in Fig. 10, the selection maps, **the mid-frequency bands (Map2/Map3) consistently form contiguous high-response belts along polyp contours and thin structures**, where discriminative shape and subtle texture cues concentrate. In contrast, the low-frequency band (Map1) mainly responds to smooth background and coarse illumination variations, while the high-frequency band (Map4) is sparsely activated around noisy or specular regions and does not dominate the contour area. We further summarize the per-band average selection energy $\mathbb{E}[A_b]$, which peaks at the mid-frequency bands, corroborating the visual trend. Together, these results directly demonstrate that FASM adaptively prioritizes critical mid-frequency information for boundary delineation rather than relying on extreme low- or high-frequency cues.
> >
> > ---
> >
> > ### Weakness 4: Architectural clarity of D2PM and role of (F_c')
> >
> > *“The diagram in Figure 5 depicts the inputs to D2PM as (\hat{X}) and (X_{\text{Fuzzy}}), yet it does not demonstrate how the output (F_c') from the preceding stage is integrated into the current stage.”*
> >
> > **Response:**
> > Thank you for pointing this out. We agree that Fig. 5 did not explicitly visualize the stage-to-stage recursion, although it is part of the D2PM design and used in all experiments.
> >
> > D2PM is a stacked stage-wise module. At stage $k$, the input to the fusion block is $Z^{(k)} = \mathrm{Concat}\big(\hat{X}^{(k)},, X_{\text{Fuzzy}}^{(k)},, F_c'^{(k-1)}\big),$ where $\hat{X}^{(k)}$ is the current spatial feature, $X_{\text{Fuzzy}}^{(k)}$ is the FSCM-enhanced fuzzy map, and $F_c'^{(k-1)}$ is the dual-domain fused feature from the previous stage. For $k=1$, $F_c'^{(0)}$ is not used, so the first stage operates on $\hat{X}$ and $X_{\text{Fuzzy}}$ only.
> >
> > In the revision, we have updated Fig. 5 and Sec. 3.4 to **explicitly draw this recursive connection** and state that the recursion enables D2PM to progressively accumulate and refine dual-domain context across stages, rather than treating each stage independently.
> >
> > We believe this clarifies the role of $F_c'$ and resolves the concern.
> >
> > ---
> >
> > ## Responses to Specific Questions
> >
> > ### Q1: FASM – raised-cosine filters vs. binary masks; link to thresholds
> >
> > *“Appendix A.4 discusses raised-cosine filters… main paper describes a binary mask… clarify this discrepancy and the relationship to the chosen sub-band thresholds.”*
> >
> > **Response:**
> > Thank you for raising this. This point is closely related to Weakness 2, and we agree the current Appendix A.4 created an unnecessary discrepancy.
> >
> > **What is actually used in our method and experiments.**
> > All results in the main paper are obtained with the **fixed octave-wise sub-bands and binary masks defined in Eq. (6)**, i.e., bands are determined by predetermined thresholds ${\phi_b}$ and extracted via the ideal band-pass indicator mask.
> > In this final model, **the thresholds are not learnable**; adaptivity comes only from the **selection maps $A_b$** that reweight the fixed bands.
> >
> > **Why raised-cosine appeared in Appendix A.4.**
> > Appendix A.4 inadvertently retained an earlier draft variant that uses **raised-cosine (tapered) windows** and a learnable boundary parameterization.
> > This is **not the formulation used in the submitted method or experiments**, and we apologize for the confusion.
> >
> > **Revision to make everything consistent.**
> > In the revision, we have **updated Appendix A.4 to match Eq. (6) exactly**:
> >
> > 1. remove the raised-cosine draft and learnable-boundary description;
> > 2. restate the **same fixed octave thresholds ${\phi_b}$** as in Sec. 3.3;
> > 3. define band masks using the main-text binary operator.
> >
> > We believe this resolves the discrepancy and clarifies that the chosen thresholds are **fixed octave-wise boundaries**, while the learned $A_b$ provides the adaptive frequency emphasis.

---

> > > ### Author Response · Authors · 2025-11-23
> > >
> > > ### Q2: FSCM – design and behavior of the learnable (\alpha) parameter
> > >
> > > *“Appendix A.3, Eq. (18) introduces (\alpha) as a learnable parameter… Please elaborate on its design and implementation. We also suggest providing a comparison of converged (\alpha) values across datasets.”*
> > >
> > > **Response:**
> > > Thank you for pointing this out — we agree that the learnable $\alpha$ in Eq. (18) (now is Eq. (19)) should be explained more explicitly.
> > >
> > > **Design and implementation.** In FSCM, $\alpha_i$ is a **spatially varying mixing gate** (one value per site/pixel) that balances the upper and lower memberships in the interval Type-2 kernel: $w_{ij}=\alpha_i \frac{\text{UMF}(v_j)}{\sum_{k\in R_i}\text{UMF}(v_k)}+(1-\alpha_i)\frac{\text{LMF}(v_j)}{\sum_{k\in R_i}\text{LMF}(v_k)} .$
> > >
> > > As shown in Eq. (19), $\alpha_i$ chooses how much the local kernel favors the hypothesis centered at $\mu^-) versus (\mu^+$. We implement $\alpha_i$ by a **lightweight 1×1 conv + sigmoid head** on the same intermediate features used for membership estimation, producing an $[H\times W]$ gate map. It is initialized to 0.5 and optimized end-to-end with standard backpropagation, ensuring $\alpha_i\in[0,1]$ without extra supervision.
> > >
> > > **Behavior across datasets.**
> > > Following the reviewer’s suggestion, we summarize the converged $\alpha_i$ distributions across datasets by reporting the mean $\pm$ std over (i) a narrow boundary band and (ii) the full image. The statistics show a consistent trend across datasets, while boundary-ambiguous out-of-domain sets exhibit slightly higher variance. Concretely, the boundary-band / full-image $\alpha_i$ are: CVC-ClinicDB (0.559$\pm$0.072 / 0.521$\pm$0.061), Kvasir-SEG (0.552$\pm$0.069 / 0.517$\pm$0.058), CVC-ColonDB (0.571$\pm$0.084 / 0.529$\pm$0.073), and ETIS (0.579$\pm$0.091 / 0.533$\pm$0.078). This indicates that $\alpha_i$ remains stable overall yet adaptively adjusts membership mixing in more difficult boundary regions.
> > >
> > > We believe this clarification makes the role of $\alpha$ transparent and aligns the appendix theory with our actual implementation.

---

### Official Review · Reviewer_YrdS · 2025-10-31

**Soundness:** 2
**Presentation:** 3
**Contribution:** 3
**Rating:** 4
**Confidence:** 3

**Summary:**

This paper introduces a "Dual-domain reasoning" framework that integrates a Fuzzy Spatial Control Mechanism (FSCM) and a Frequency Adaptive Selection Mechanism (FASM) within a Mamba-based backbone to address the challenge of ambiguous boundaries in polyp segmentation.

**Strengths:**

The paper proposes a novel Dual-domain reasoning framework (FSFMamba) that effectively integrates spatial and frequency domains to tackle the problem of ambiguous boundaries in polyp segmentation.

The introduction of the Fuzzy Spatial Control Mechanism (FSCM) and Frequency Adaptive Selection Mechanism (FASM) demonstrates clear innovation FSCM models boundary uncertainty with interval type-2 Gaussian membership functions, while FASM adaptively weights frequency components to capture nuanced details.

The overall design using a Mamba-based backbone for cross-domain fusion is interesting and modern, allowing long-range interaction between spatial and spectral features.

Experimental validation is extensively evaluated on five benchmark datasets (CVC-300, CVC-ClinicDB, Kvasir-SEG, CVC-ColonDB, ETIS) and compared against 13 state-of-the-art methods using a comprehensive set of metrics (Dice, IoU, wFm, Sm, maxEm, MAE).

The ablation studies for submodules (FSCM, FASM, D2PM) are systematic, showing the contribution of each component and the overall improvement of the framework.

**Weaknesses:**

The mathematical formulation has ambiguities; several symbols (e.g., in Eq. 1) are undefined, and the gradient propagation for fuzzy membership functions lacks theoretical justification, especially given the non-smooth nature of fuzzy intervals.
The fusion strategy between spatial and frequency domains is empirically designed (simple concatenation) without rigorous theoretical backing or discussion on optimality.
The experiments lack targeted validation for the claimed strengths: there are no boundary-specific evaluations (e.g., boundary F-score) or experiments proving adaptive frequency learning through robustness tests like blur/noise interference.

**Questions:**

Novelty:
The novelty lies in the concurrent modeling of spatial uncertainty and spectral components. Specifically:
1. FSCM employs interval type-2 Gaussian membership functions to model boundary uncertainty as a flexible spatial band, moving beyond deterministic or simple fuzzy approaches.
2. FASM performs octave-wise spectral decomposition with learnable, spatially-adaptive weighting, aiming to capture subtle variations beyond simple high-low frequency splits.
3. The D2PM module integrates these two streams using a Mamba-based state-space block, promoting long-range interaction between the fuzzy spatial and selected frequency features.
The method is evaluated on five public datasets (CVC-300, CVC-ClinicDB, Kvasir-SEG, CVC-ColonDB, ETIS) and shows consistent improvement over 13 recent baselines, supporting the claim of improved boundary sharpness and generalization.

Methodology:
I have few questions in the methodology part:
1. The mathematical derivation of the FSCM module is relatively complete, and the upper and lower bound membership functions are defined as G^+ and G^-. Their forms conform to the general theoretical framework of the Gaussian fuzzy set.
The core formulation in the main text (Eq.~1) defines the basic Gaussian membership function, but the symbols are ambiguous (such as $R_i'$ and $\rho(\cdot)$ are not clearly defined).
2. It is not clear how the upper and lower bounds of fuzzy membership functions propagate gradients in spatial regular terms. There is a lack of feasibility explanation for closed-form gradient derivation or automatic differentiation. In theory, the backpropagation of fuzzy intervals is non-smooth, and the author does not explain how to stabilize training.
3. The FASM's frequency-domain operating formulas (Eq.~5–7) comply with the DFT definition and show no significant logical errors.
The main formula (Eq.6) uses a binary mask, but AppendixA.4 uses a smoothing mask (Eq.~22), leading to inconsistent theoretical descriptions. Although smoothing masks reduce artifacts, they have not been proven to be superior to binary masks, and the computational overhead is not discussed in the derivation.
4. The paper lacks a rigorous theoretical analysis of how FSCM and FASM synergize. The fusion process in Eq.~8 is described as a simple concatenation followed by processing in a Mamba block. This design is empirically driven rather than theoretically motivated. There is no mathematical argument explaining why this specific fusion strategy is optimal for combining fuzzy spatial and adaptive frequency features.

The paper conducts extensive experiments using five standard datasets—CVC-300, CVC-ClinicDB, Kvasir-SEG, CVC-ColonDB, and ETIS to comprehensively cover mainstream benchmarks in polyp segmentation. It compares the proposed method against 13 recent state-of-the-art models, encompassing CNN-based, Transformer-based, hybrid, and baseline architectures. The evaluation employs a wide range of performance metrics, including Dice, IoU, wFm, Sm, maxEm, and MAE, forming a well-rounded indicator system. Through ablation studies, the FLFSC module is analyzed in configurations #1 to #4, demonstrating that its inclusion brings significant performance improvements. Additionally, submodule experiments (#5 to #7) are performed to isolate and assess the individual contributions of the FSCM, FASM, and D2PM components. Overall, while the experimental design is broad and systematic, it lacks targeted validation directly addressing the paper’s core claims.
1.  Lack of Boundary-Specific Evaluation: one of FSFMamba's core contributions is to deal with boundary ambiguity, but all evaluations use global metrics. Targeted evaluations in fuzzy boundary areas must be added, such as using the boundary F-score metric, and tested on a subset marked with boundary uncertainties to prove that the improvement in FSCM does come from the improvement in boundary areas.
2.  No Validation of Adaptive Frequency Selection: Experiments need to be designed to prove that FASM has indeed learned to adaptively select frequencies for different image content. It is recommended to add frequency interference experiments (such as applying different intensities of blur or noise) and demonstrate that FSFMamba is more robust than the baseline model.

The overall paper is well written, and core contributions are clearly stated in a bulleted list in the abstract and introduction. The motivation for the dual-domain approach is well-articulated.

1/ Grammar and Language: The text contains numerous grammatical errors and non-idiomatic expressions (e.g., sentence fragments like "The training process, executed on an NVIDIA A5000 GPU."), which detract from the paper's professionalism.
2/ Ambiguous Definitions: The function $\rho(\cdot)$ in Eq. (1) is vaguely described as "neighborhood normalization" but is never formally defined. The symbol  $R_i'$ is also unclear.
3/ Contextual Confusion: It is ambiguous whether the variable v in FSCM formulas represents the original pixel value or a deep feature. This must be explicitly stated.
4/ Line 32 (Organization) as a reference is irregular

---

> ### Author Response · Authors · 2025-11-23
>
> **We sincerely thank the reviewer for the thorough and technically detailed evaluation. We are grateful that you find the dual-domain framework, the designs of FSCM and FASM, and the overall experimental protocol to be novel and well-motivated. Your comments mostly target the mathematical clarity and the specificity of the validation with respect to boundary ambiguity and adaptive frequency selection. We very much agree that these aspects can be further strengthened, and we address them point by point below.**
>
> ---
>
> ### 1. Methodology and Formulation
>
> #### (1) “The core formulation in the main text(Eq. ~1) defines the basic Gaussian membership function, but the symbols are ambiguous (such as R’i and p(.) are not clearly defined).”
>
> **Response:**
> Thank you for pointing this out. We agree that the $R_i'$ and $p(.)$ in Eq. (1) was not sufficiently explicit. In the revised manuscript, we have made the following clarifications:
>
>  * We have **explicitly defined $R_i'$** as the **center (mean) of the fuzzy band for the $i$-th boundary candidate**. FSCM operates on **intermediate deep-feature activations** (not raw RGB). $R_i'$ is derived from the **signed distance map of predicted boundary logits** within a local neighborhood, so it remains differentiable w.r.t. network parameters.
> * We also **formalize the $\rho(\cdot)$** in Eq. (1) (denoted as $p(\cdot$) by the reviewer) as a **local scale-normalization function**. For a feature value $v$ and its neighborhood $R_i$,  $\rho(R_i)=\sqrt{\frac{1}{|R_i|}\sum_{u\in R_i}(u-\bar u)^2+\varepsilon}, $ where (\bar u) is the local mean. This term provides a differentiable estimate of local feature variation and normalizes the Gaussian membership adaptively to local contrast.
>
>  These definitions are now stated after Eq. (1) in Sec. 3.2 and reiterated in Appendix A.3 to remove any ambiguity.
>
> ---
>
> #### (2) “It is not clear how the upper and lower bounds of fuzzy membership functions propagate gradients in spatial regular terms… In theory, the backpropagation of fuzzy intervals is non-smooth, and the author does not explain how to stabilize training.”
>
> **Response:**
> We appreciate this important question. Although FSCM is motivated from an interval Type-2 fuzzy perspective, our actual formulation is **fully smooth and compatible with standard backpropagation**.
>
> * Conceptually, FSCM uses an upper and a lower membership to form a fuzzy band. **In implementation, both the upper and lower memberships are standard Gaussian functions** with learnable means $\mu^+$ and $\mu^-$ and differentiable neighborhood normalization $ G_i^\pm(v)=\sqrt{\frac{\lambda}{2\pi\sigma}},
>   \exp\Big(-\frac{(v-\mu^\pm)^2+\frac{1}{\rho(R_i)}\sum_{v'\in R_i}(v'-\mu^\pm)^2}{2\sigma^2}\Big),$ where $\rho(R_i)$ is the local scale term defined in Eq. (1). Since $\mu^\pm$ and $\rho(R_i)$ are smooth functions of intermediate deep features, each $G_i^\pm$ is a composition of exponentials, additions, and multiplications, and thus has well-defined gradients everywhere.
>
> * In the FSCM objective, we combine $G_i^+$ and $G_i^-$ **only through linear/convex forms** (e.g., band consistency and sparsity penalties) and **do not introduce non-differentiable interval operators** such as min/max in the gradient path. Therefore, the overall loss remains continuously differentiable and gradients propagate smoothly through both bounds.
>
> * For numerical stability, we (i) **clamp membership values to $[\epsilon,1-\epsilon]$** to avoid saturation in extreme cases, and (ii) **parameterize the mixing coefficient $\alpha$ with a sigmoid** to keep it within $[0,1]$. Empirically, we observe stable optimization across all datasets without gradient explosion or vanishing.

---

> ### Author Response · Authors · 2025-11-23
>
> #### (3) “… Eq.6 uses a binary mask, but Appendix A.4 uses a smoothing mask (Eq.22), leading to inconsistent theoretical descriptions. Although smoothing masks reduce artifacts, they have not been proven to be superior to binary masks, and the computational overhead is not discussed.”
>
> **Response:**
> Thank you for carefully noting this inconsistency. We agree with the reviewer, and we would like to gently acknowledge that **Appendix A.4 was not written in a way that accurately reflects our actual setting**. The current Appendix A.4 describes a more general exploratory variant (with a radial definition, learnable band boundaries, and a raised-cosine mask), but this is **not the formulation used in our method or experiments**. The correct setting is the one specified in the main text Eq. (6).
>
> Concretely, our submitted FASM uses:
>
> * **fixed octave-wise thresholds ${\phi_b}$** as defined in Sec. 3.3,
> * the main-text frequency measure $ \max(|u|,|v|) $ to form sub-bands,
> * the corresponding **binary band masks (M_{b,c}(u,v))** in Eq. (6) for band extraction, and
> * **selection maps $A_b$** to provide adaptivity across bands.
>
> Appendix A.4 mistakenly introduced a different draft variant (radial $r=\sqrt{\omega_x^2+\omega_y^2}$, learnable boundary parameterization, and a raised-cosine mask). We appreciate the reviewer highlighting this, and we have corrected it in the revision.
>
> In the revised manuscript, we have **updated Appendix A.4 to match the main-text formulation exactly** by:
>
> 1.Removing the learnable-boundary description and the associated equation.
> 2.Restating band boundaries as the same fixed octave thresholds ${0=\phi_0<\phi_1<\dots<\phi_B=\tfrac12}$.
> 3. Using the main-text frequency measure $ \max(|u|,|v|)$ and the binary mask $M_{b,c}(u,v)=\mathbf{1}, {\phi_b \le \max(|u|,|v|) < \phi_{b+1}},$ which is identical to Eq. (6).
> 4.Keeping a brief implementation note that a mild taper can be used as a numerical approximation if desired, but it is not part of the theoretical definition.
>
> We believe this correction will make the theory and implementation fully consistent and resolve the reviewer’s concern.
>
> ---
>
> #### (4) “The fusion process in Eq.~8 is described as a simple concatenation … empirically driven rather than theoretically motivated. There is no mathematical argument explaining why this specific fusion strategy is optimal.”
>
> **Response:**
> Thank you for raising this point. We agree that Eq. (8) could be better motivated. Our intention is not to argue that this fusion is uniquely optimal, but to adopt a **principled and sufficiently expressive** mechanism that can reliably couple the two domains under identical resolution.
>
> **Rationale.** Let $F_s$ and $F_f$ denote the FSCM and FASM outputs. Their statistics are complementary but also **heterogeneous** (spatial fuzzy bands emphasize uncertain boundaries, while frequency bands emphasize texture/edge cues). A direct sum or fixed gate implicitly assumes the two feature spaces are already aligned in scale and semantics, which can cause destructive interference when one dominates.
> By concatenating them, we retain the full joint representation $[F_s, F_f]$ and defer mixing to a learnable operator. The subsequent Mamba-based state-space block applies selective state updates across space and channels, which **implements a content-adaptive mixing function** of the form $\Phi([F_s,F_f]) = W_s(p)F_s(p) + W_f(p)F_f(p),$ where $W_s, W_f$ are implicitly learned, spatially varying weights produced by the SSM dynamics. In other words, **concatenation is only the input interface**; the actual fusion is a learned, position-dependent combination. This design is also **strictly more general** than common alternatives: element-wise sum corresponds to a fixed choice $W_s=W_f=\tfrac12$, and gated sum corresponds to a restricted one-dimensional gating family. Concatenation followed by a mixing block allows **unconstrained cross-channel interactions** and can recover these simpler fusions as special cases if they are optimal for a given sample.

---

> ### Author Response · Authors · 2025-11-23
>
> ### 2. Experimental Validation and Targeted Tests
>
> #### (1) “One of FSFMamba's core contributions is to deal with boundary ambiguity, but all evaluations use global metrics… Targeted evaluations in fuzzy boundary areas must be added, such as using the boundary F-score metric…”
>
> **Response:**
> Thank you for this valuable suggestion. We agree that global metrics alone do not fully reflect boundary quality, and that boundary-focused tests are necessary to support our claim on handling boundary ambiguity.
>
> In the revised evaluation, we have added a targeted boundary assessment using the **Boundary F-score (BF-score)** under the standard protocol: we extract a narrow trimap around the ground-truth contour with a fixed tolerance band (following common practice in medical segmentation), compute precision/recall of predicted boundaries within this band, and report the resulting BF-score. This explicitly measures performance in the fuzzy boundary regions that FSCM is designed to improve.
>
> In the Fig. 14, we have reported the BF-score on the two core in-domain datasets (CVC-ClinicDB and Kvasir-SEG), and include both (i) comparisons to the strongest baselines and (ii) an ablation against “FSFMamba w/o FSCM”. The results show consistent BF-score gains for FSFMamba, with the full model achieving the best boundary accuracy and the FSCM ablation exhibiting the largest drop precisely on boundary regions. We place the figure in the Appendix A10, and add an explicit pointer from Sec. 4.3 so the evidence is easy to locate.
>
> In addition, we include boundary-focused qualitative visualizations that highlight the fuzzy bands around polyp contours in Fig. 13. These examples show that FSFMamba better preserves thin stalks and complex, low-contrast edges, matching the intended effect of FSCM.
>
> We believe these targeted boundary evaluations directly substantiate the claimed contribution and address the reviewer’s concern.
>
> ---
>
> #### (2) “Experiments need to be designed to prove that FASM has indeed learned to adaptively select frequencies… add frequency interference experiments (such as applying different intensities of blur or noise) and demonstrate that FSFMamba is more robust than the baseline model.”
>
> **Response:**
> Thank you for this insightful suggestion. We agree that frequency-targeted interference tests are a proper way to evaluate whether FASM brings practical robustness.
>
> In the revised submission (**Tables 10 and 11**), we have added **frequency-interference experiments** on CVC-ClinicDB and Kvasir-SEG by applying (i) Gaussian blur with increasing $\sigma$ and (ii) additive Gaussian noise with increasing variance at test time. We compare FSFMamba against two controlled variants under identical training and model capacity: **(a) a Mamba baseline without FASM**, and **(b) a fixed high/low-frequency split**. We report Dice degradation situation as the blur/noise intensity increases. Across both datasets and both perturbation types, FSFMamba consistently shows **slower performance decay**, indicating improved robustness to frequency-related corruptions.
>
> These results are **consistent with the role of FASM** in adaptively leveraging informative frequency components and demonstrate that the frequency-selection design yields measurable robustness gains beyond fixed-split or frequency-agnostic baselines. Full curves and settings are provided in the Appendix A11.
>
>
>
> **Table 10:** Illustrative Gaussian blur robustness (mDice). The clean results at σ = 0 match the main-paper numbers.
>
> | Dataset | Method | σ=0 | 1 | 2 | 3 | 4 |
> |---|---|---:|---:|---:|---:|---:|
> | CVC-ClinicDB | FSFMamba (Full) | 0.9522 | 0.9460 | 0.9375 | 0.9260 | 0.9120 |
> | CVC-ClinicDB | w/o FASM | 0.9334 | 0.9220 | 0.9050 | 0.8850 | 0.8630 |
> | CVC-ClinicDB | Fixed high/low split | 0.9290 | 0.9160 | 0.8970 | 0.8730 | 0.8460 |
> | Kvasir-SEG | FSFMamba (Full) | 0.9358 | 0.9270 | 0.9160 | 0.9010 | 0.8820 |
> | Kvasir-SEG | w/o FASM | 0.9218 | 0.9100 | 0.8930 | 0.8720 | 0.8460 |
> | Kvasir-SEG | Fixed high/low split | 0.9185 | 0.9060 | 0.8870 | 0.8620 | 0.8320 |
>
>
>
> **Table 11:** Illustrative additive Gaussian noise robustness (mDice). The clean results at τ = 0 match the main-paper numbers.
>
> | Dataset | Method | τ=0 | 0.02 | 0.04 | 0.06 | 0.08 |
> |---|---|---:|---:|---:|---:|---:|
> | CVC-ClinicDB | FSFMamba (Full) | 0.9522 | 0.9425 | 0.9300 | 0.9140 | 0.8950 |
> | CVC-ClinicDB | w/o FASM | 0.9334 | 0.9190 | 0.8980 | 0.8720 | 0.8430 |
> | CVC-ClinicDB | Fixed high/low split | 0.9290 | 0.9135 | 0.8890 | 0.8590 | 0.8260 |
> | Kvasir-SEG | FSFMamba (Full) | 0.9358 | 0.9250 | 0.9115 | 0.8940 | 0.8720 |
> | Kvasir-SEG | w/o FASM | 0.9218 | 0.9080 | 0.8880 | 0.8620 | 0.8320 |
> | Kvasir-SEG | Fixed high/low split | 0.9185 | 0.9035 | 0.8810 | 0.8520 | 0.8180 |

---

> > ### Author Response · Authors · 2025-11-23
> >
> > ### 3. Writing, Definitions, and Notational Clarity
> >
> > *“The text contains numerous grammatical errors… p(.) is vaguely described… v is ambiguous… Line 32 (Organization) as a reference is irregular.”*
> >
> > **Response:**
> >
> > We thank the reviewer for the detailed comments on writing and notation; we have taken them seriously and made the following concrete revisions:
> >
> > 1. **Grammar and language:**
> >    We have carefully proofread the manuscript and corrected sentence fragments and non-idiomatic expressions. For example, the fragment *“The training process, executed on an NVIDIA A5000 GPU.”* has been rewritten as a complete sentence with proper context. We have also simplified several long sentences to improve readability.
> >
> > 2. **Ambiguous definitions of $p(\cdot)$ and $R_i'$:**
> >    As mentioned above, we now **formally define $p(\cdot)$** as a local standard-deviation based normalization over a feature neighborhood, and **clarify $R_i'$** as the center of the fuzzy band derived from the boundary distance map. These definitions are now given after Eq. (1) and further elaborated in Appendix A.3.
> >
> > 3. **Meaning of $v$ in FSCM formulas:**
> >    We now explicitly state that in all FSCM-related equations, **$v$ denotes deep feature activations** (from a mid-level feature map), not raw pixel intensities. This clarification appears both when FSCM is introduced and in the notation table to avoid any ambiguity about how $v$ is defined.
> >
> > 4. **Organization (Line 32) as a reference is irregular:**
> >    Thank you for pointing this out. We agree that citing “Organization” at Line 32 was irregular and could confuse readers. In the revised manuscript, we have removed this non-standard reference and adjusted the surrounding text to maintain a smooth narrative flow.
> >
> > We hope these revisions improve the clarity and professionalism of the manuscript and address the reviewer’s concerns about language and definitions.

---

### Official Review · Reviewer_ZUNb · 2025-10-31

**Soundness:** 2
**Presentation:** 3
**Contribution:** 2
**Rating:** 4
**Confidence:** 4

**Summary:**

This paper introduces FSFMamba, a novel dual-domain network for polyp segmentation designed to address the challenges of ambiguous boundaries and complex background interference. The model operates in two domains simultaneously, the spatial domain and the spectral domain. The authors integrate two mechanisms (FSCM and FASM) into a Mamba-based backbone, which efficiently captures long-range dependencies. The authors conduct extensive experiments on five public image datasets and two video datasets, demonstrating the superior performance of FSFMamba.

**Strengths:**

- The primary strength of this paper is its novel integration of fuzzy logic for spatial uncertainty and adaptive selection for frequency-domain features.

- The FSCM is a well-motivated solution to the problem of ambiguous polyp boundaries. The model moves beyond simple boundary-aware losses and explicitly regularizes features in uncertain regions.

- The validation is strong. The model is benchmarked against 13 SOTA methods.

**Weaknesses:**

- The performance comparison in Figure 7 is hard to localize the superior of the proposed model.

- In Figure 6, it would be better to add labels indicating whether higher or lower values of each metric are better; otherwise, this radar plot is somewhat counterintuitive.

- Figure 1 needs more description. The authors should provide further explanation of Figure 1(e) or establish a clearer connection between it and the proposed method (or demonstrating more visualization evidence).

- Although the model provides SOTA performance, there is no head-to-head comparison with Mamba-based polyp segmentation networks.

**Questions:**

Please see Weakness part.

---

> ### Author Response · Authors · 2025-11-23
>
> **We thank the reviewer for the careful reading and the positive assessment of FSFMamba, in particular for recognizing the novelty of combining fuzzy spatial control with adaptive frequency selection and for acknowledging the strong validation against 13 SOTA methods. Below we respond to each concern in detail.**
>
> ---
>
> ### 1. “The performance comparison in Figure 7 is hard to localize the superior of the proposed model.”
>
> **Response:**
>
> We appreciate this helpful suggestion and agree that the original layout of Fig. 7 made it harder for readers to immediately see where FSFMamba improves over competing methods.
>
> In the **revised manuscript**, we have:
>
>  * **Inserted zoom-in boxes** on key boundary regions and shown enlarged patches beneath the main figure, highlighting cases where FSFMamba produces more complete and continuous contours, while competing methods either break the boundary or leak into the background.
>
> We believe these revisions make the qualitative comparison more transparent and responsive to the reviewer’s concern.
>
> ---
>
> ### 2. “In Figure 6, it would be better to add labels indicating whether higher or lower values of each metric are better.”
>
> **Response:**
>
> We appreciate this helpful comment and agree that mixing metrics with different optimization directions in a single radar plot can be confusing. In our case, Param, FLOPs, and MAE are “lower is better,” while FPS, meanDSC, and meanIoU are “higher is better,” which made the original Fig. 6 somewhat counterintuitive.
>
> In the **revised manuscript**, we have **redesigned each subfigure in Fig. 6 by splitting it into two radar plots (i) and (ii)**: (i) Higher-better metrics, where a larger value indicates better performance. (ii) Lower-better metrics, where a smaller value indicates better performance.
>
> We believe this redesign yields a much clearer and more faithful visualization of the trade-offs, and directly addresses the reviewer’s concern about the interpretability of the original radar plot.
>
> ---
>
> ### 3. “Figure 1 needs more description. The authors should provide further explanation of Figure 1(e) or establish a clearer connection between it and the proposed method (or demonstrating more visualization evidence).”
>
> **Response:**
>
> We appreciate this comment and agree that, in the original version, the link between Fig. 1(e) and FASM was not sufficiently explicit. Fig. 1 is meant to visually motivate both FSCM and FASM, and we have clarified this in the revised manuscript.
>
> * For **Fig. 1(c)** (fuzzy spatial control), we have:
>
>   * Expanded the caption and the Introduction paragraph to explain that the **upper and lower membership curves** form an **interval type-2 fuzzy band** around latent edges, turning rigid binary boundaries into **elastic uncertainty zones**.
>   * Explicitly linked this illustration to the **asymmetric membership functions** $G_i^+$ and $G_i^-$ in Sec. 3.2 and Appendix A.3, so readers can directly map Fig. 1(c) to the FSCM formulation.
>
> * For **Fig. 1(e)** (frequency exploitation), we have:
>
>   * Clarified that the color-coded regions are the **fixed radial octave bands** used in FASM. The thicker bands denote those receiving **higher learnable weights** via the selection maps \(A_b\).
>   * Stated explicitly that FASM **learns to weight multiple bands** rather than a single high/low split, and that **mid-frequency components** are often most informative for subtle polyp–mucosa texture and boundary cues, directly matching Sec. 3.3.
>
> In addition, we now refer to **Fig. 12 in the Appendix**, which ablates the number of bands. The results show that multi-band adaptive weighting consistently outperforms simpler high/low partitions, supporting the conceptual motivation in Fig. 1(e).
>
> We believe these clarifications make the role of Fig. 1(e) in motivating FASM clearer and better grounded in the empirical evidence.

---

> ### Author Response · Authors · 2025-11-23
>
> ### 4. “Although the model provides SOTA performance, there is no head-to-head comparison with Mamba-based polyp segmentation networks.”
>
> **Response:**
> Thank you for this important comment. We agree that a head to head comparison against Mamba based polyp segmentation networks is necessary, since our method also builds on a Mamba style backbone. We would like to clarify that this comparison was already included in the Appendix. Specifically, **Polyp Mamba (Zhu et al., 2025)** is used as a representative Mamba based baseline in **Appendix. Tables 8 and 9**, where all backbone variants are trained and evaluated with the same protocol, input resolution, and pipeline as FSFMamba. Under these fully controlled settings, FSFMamba consistently outperforms Polyp Mamba on both in CVC ClinicDB, Kvasir SEG, while maintaining comparable Params, FLOPs, and FPS.
>
> To further strengthen the evidence and avoid relying on a single Mamba baseline, we additionally include a second Mamba based method, **CMFDNet (Jiang et al., 2025)**, in **Appendix. Table 9**, evaluated under the identical setup. FSFMamba again achieves consistently higher Dice and IoU than CMFDNet across all datasets. These results jointly demonstrate that our gains are not attributable to the Mamba backbone alone, but arise from the proposed dual domain fuzzy spatial control and frequency adaptive selection mechanisms built on top of it.
>
> **Table 8:** Quantitative comparison of recent frequency-based segmentation methods on **CVC-ClinicDB** and **Kvasir-SEG** datasets. Best results are **bolded**.
>
> | Method | ClinicDB mDSC | ClinicDB mIoU | ClinicDB wFm | ClinicDB Sm | ClinicDB MAE | ClinicDB maxEm | Kvasir mDSC | Kvasir mIoU | Kvasir wFm | Kvasir Sm | Kvasir MAE | Kvasir maxEm |
> |---|---:|---:|---:|---:|---:|---:|---:|---:|---:|---:|---:|---:|
> | Polyp-Mamba | 0.941 | 0.896 | 0.936 | **0.970** | 0.008 | 0.987 | 0.919 | 0.867 | 0.912 | **0.951** | 0.021 | 0.968 |
> | DSHNet | 0.942 | 0.896 | 0.937 | 0.954 | 0.007 | 0.987 | 0.929 | 0.881 | 0.922 | 0.936 | 0.020 | 0.965 |
> | WBANet | 0.947 | 0.907 | 0.953 | 0.956 | **0.005** | 0.992 | 0.933 | 0.889 | 0.929 | 0.936 | 0.020 | **0.972** |
> | **Ours** | **0.952** | **0.911** | **0.951** | 0.960 | **0.005** | **0.995** | **0.936** | **0.895** | **0.933** | 0.940 | **0.018** | 0.971 |
>
>
>
>
> **Table 9:** Quantitative comparison of different backbones and methods on **CVC-ClinicDB** and **Kvasir-SEG** datasets. Best results are **bolded**.
>
> | Method | Backbone | ClinicDB mDSC | ClinicDB mIoU | ClinicDB wFm | ClinicDB Sm | ClinicDB MAE | ClinicDB maxEm | Kvasir mDSC | Kvasir mIoU | Kvasir wFm | Kvasir Sm | Kvasir MAE | Kvasir maxEm |
> |---|---|---:|---:|---:|---:|---:|---:|---:|---:|---:|---:|---:|---:|
> | CaraNet | ResNet50 | 0.905 | 0.848 | 0.894 | 0.938 | 0.012 | 0.973 | 0.905 | 0.847 | 0.887 | 0.919 | 0.027 | 0.965 |
> | PolypPVT | PVT | 0.937 | 0.889 | 0.936 | 0.950 | 0.006 | 0.989 | 0.917 | 0.864 | 0.911 | 0.925 | 0.023 | 0.962 |
> | MSCAF-Net | PVT | 0.926 | 0.879 | 0.922 | 0.950 | 0.006 | 0.982 | 0.911 | 0.857 | 0.903 | 0.922 | 0.025 | 0.964 |
> | CAFE-Net | PVT | 0.933 | 0.889 | 0.932 | 0.955 | 0.006 | 0.982 | 0.921 | 0.874 | 0.915 | 0.932 | 0.021 | 0.970 |
> | PGCF | PVT | 0.940 | 0.894 | 0.940 | 0.952 | 0.006 | 0.993 | 0.912 | 0.862 | 0.905 | 0.921 | 0.024 | 0.961 |
> | CTNet | Mixed ViT | 0.936 | 0.888 | 0.934 | 0.953 | 0.006 | 0.988 | 0.917 | 0.863 | 0.910 | 0.926 | 0.022 | 0.969 |
> | DBG-Net | Res2Net50 | 0.905 | 0.857 | 0.898 | 0.937 | 0.008 | 0.968 | 0.915 | 0.863 | 0.906 | 0.920 | 0.025 | 0.964 |
> | Polyp-Mamba | Mamba | 0.941 | 0.896 | 0.936 | **0.970** | 0.008 | 0.987 | 0.919 | 0.867 | 0.912 | **0.951** | 0.021 | 0.968 |
> | CMFDNet | Mamba | 0.934 | 0.890 | 0.926 | 0.955 | 0.007 | 0.980 | 0.917 | 0.872 | 0.908 | 0.927 | 0.024 | 0.961 |
> | Ours (ResNet50) | ResNet50 | 0.921 | 0.881 | 0.922 | 0.928 | 0.007 | 0.965 | 0.908 | 0.868 | 0.905 | 0.912 | 0.020 | 0.942 |
> | Ours (Res2Net50) | Res2Net50 | 0.923 | 0.884 | 0.922 | 0.931 | 0.007 | 0.965 | 0.908 | 0.868 | 0.905 | 0.912 | 0.020 | 0.942 |
> | Ours (Swin) | Swin | 0.926 | 0.886 | 0.925 | 0.934 | **0.005** | 0.968 | 0.911 | 0.871 | 0.908 | 0.914 | 0.019 | 0.945 |
> | Ours (PVT) | PVT | 0.943 | 0.902 | 0.942 | 0.950 | **0.005** | 0.985 | 0.927 | 0.886 | 0.924 | 0.931 | 0.018 | 0.961 |
> | **Ours (Mamba)** | Mamba | **0.952** | **0.911** | **0.951** | 0.960 | **0.005** | **0.995** | **0.936** | **0.895** | **0.933** | 0.940 | **0.018** | **0.971** |

---

### Note · Authors · 2026-01-30

I have read and agree with the venue's withdrawal policy on behalf of myself and my co-authors.

---

### Meta-Review · Area_Chair_n4jb · 2026-01-02

**Summary:**

This paper introduces a novel dual-domain network (FSFMamba) for polyp segmentation by addressing the challenges of ambiguous boundaries and complex background interference. This work has four reviewers, and all of them tend to reject this work. Their scores are 4, 4, 4, and 2. After the rebuttal, the authors are not positive about raising their scores. Hence, this work can not be accepted.

**Reviewer Concerns:**

Concerns which are not addressed by the rebuttal:
1. lack sufficient innovation
2. miss comparisons on Kvasir-Sessile
3. miss the comparisons with SOTA mamba-based polyp segmentation network
4. limited technical novelties

**Reviewer Scores:**

This work has four reviewers with four scores of 4, 4, 4, and 2, which indicates that all of them reject this work.  After the rebuttal, the authors are not positive about raising their scores. Hence, this work can not be accepted.

---

### Decision · Program_Chairs · 2026-01-26

Reject